# A nested multi-scale system implemented in the large-eddy simulation model PALM model system 6.0

Antti Hellsten[1], Klaus Ketelsen[2], Matthias Sühring[3], Mikko Auvinen[1], Björn Maronga[3,6],
Christoph Knigge[4], Fotios Barmpas[5], Georgios Tsegas[5], Nicolas Moussiopoulos[5], and Siegfried Raasch[3]

[1]Finnish Meteorological Institute, P.O.Box 503, FI-00101, Helsinki, Finland
[2]Software Consultant, Beethovenstr. 29A, 12247 Berlin, Germany
[3]Leibniz University Hannover, Institute of Meteorology and Climatology, Herrenhäuser Strasse 2, 30419 Hannover, Germany
[4]Deutscher Wetterdienst, Frankfurter Straße 135, D-63067 Offenbach, Germany
[5]Aristotle University Thessaloniki, P.O.Box 483, GR-54124, Thessaloniki, Greece
[6]University of Bergen, Geophysical Institute, Postboks 7803, 5020 Bergen, Norway

**Correspondence:** Antti Hellsten (antti.hellsten@fmi.fi)

**Abstract.**

Large-eddy simulation (LES) provides a physically sound approach to study complex turbulent processes within the atmospheric boundary layer including urban boundary layer flows. However, such flow problems often involve a large separation of turbulent scales, requiring a large computational domain and very high grid resolution near the surface features, leading to prohibitive computational costs. To overcome this problem, an online LES-LES nesting scheme is implemented into the PALM model system 6.0. The hereby documented and evaluated nesting method is capable of supporting multiple child domains which can be nested within their parent domain either in a parallel or recursively cascading configuration. The nesting system is evaluated by simulating first a purely convective boundary layer flow system and then three different neutrally-stratified flow scenarios with increasing order of topographic complexity. The results of the nested runs are compared with corresponding non-nested high- and low-resolution results. The results reveal that the solution accuracy within the high-resolution nest domain is clearly improved as the solutions approach the non-nested high-resolution reference results. In obstacle-resolving LES, the two-way coupling becomes problematic as anterpolation introduces a regional discrepancy within the obstacle canopy of the parent domain. This is remedied by introducing canopy-restricted anterpolation where the operation is only performed above the obstacle canopy. The test simulations make evident that this approach is the most suitable coupling strategy for obstacle-resolving LES. The performed simulations testify that nesting can reduce the CPU time up to 80% compared to the fine-resolution reference runs while the computational overhead from the nesting operations remained below 16% for the two-way coupling approach and significantly less for the one-way alternative.

## 1 Introduction

Large-eddy simulation (LES) has been used for basic research of atmospheric boundary layer (ABL) phenomena using idealized model setups for decades. Now, it is becoming an important method in applied research on realistic, very detailed and complicated flow systems such as urban ABL problems, e.g. (Britter and Hanna, 2003; Tseng et al., 2006; Bou-Zeid et al.,

2009; Tominaga and Stathopoulos, 2013; Giometto et al., 2016; Buccolieri and Hang, 2019; Auvinen et al., 2020a). Until the recent years, there were no ABL LES models capable of modelling detailed surface structures, such as buildings or steep complex terrain shapes in ABL. Nowadays, it is possible to carry out LES for complex built areas (e.g., Letzel et al., 2008),

but this is still limited to relatively small areas because of the high spatial resolution requirement. Concerning urban LES, Xie and Castro (2006) have shown that at least from 15 to 20 grid nodes are needed across street canyons to satisfactorily resolve the most important turbulent structures within the canyons. This requirement typically leads to grid spacings on the order of 1 m. However, the vertical extent of the LES domain should scale with the ABL height, and the horizontal size should span over several ABL heights in order to capture the ABL-scale turbulent structures (de Roode et al., 2004; Fishpool et al., 2009;

Chung and McKeon, 2010; Auvinen et al., 2020a). To adequately capture processes on the street-scale and to simultaneously capture large ABL-scale turbulence, sufficiently large model domains at small grid sizes are required, posing high demands on the computational resources in terms of CPU time and memory. Moreover, the uncertainty related to the lateral boundary conditions usually decreases as the domain is made larger.

Many conventional continuum-based numerical solution methods (e.g. finite-element and finite-volume methods) allow vari-

able resolution so that the resolution can be concentrated to the area of principal interest and relaxed elsewhere. However, only unstructured grid systems allow to take full advantage of spatially variable resolution. Many general-purpose computational fluid dynamics packages offer unstructured grid systems, but according to our experience such solvers are usually computationally decidedly less efficient than ABL-tailored LES models, such as PALM (Raasch and Schröter, 2001; Maronga et al., 2015, 2020), the Weather Research and Forecasting Model (WRF) (Skamarock et al., 2008) with its LES option and the Dutch

Atmospheric Large-Eddy Simulation (DALES) (Heus et al., 2010) that are based on structured orthogonal grid system with constant horizontal resolution. The model nesting approach can be exploited to further speed up ABL LES models or to allow larger domain sizes without compromising the resolution in the area of primary interest. The authors acknowledge that alternative solution approaches, such as Lattice Boltzmann Method (e.g., Aidun and Clausen, 2010; Ahmad et al., 2017), also offer strategies for computational efficiency improvements. The presented nesting approach is primarily relevant for the family of

structured, finite-difference Navier-Stokes solvers.

The idea of grid nesting is to simultaneously run a series of two or more LES model domains with different spatial extents and grid resolutions. In this implementation the outermost and coarsest-resolution LES domain (termed *root* domain henceforth), which acts as a *parent* to its *child* domains, obtains its boundary conditions in a conventional manner, whereas the nested LES domain (child) always obtains its boundary condition from its respective parent domain through interpolation. In one-

way coupled nesting only the children obtain information from their parents. In such a coupling strategy, the instantaneous child and parent solutions can deviate within the volume of the nest. If a stronger binding between the solutions is desired, the child solution needs to be incorporated into the parent solution. This is achieved in two-way coupled nesting, where the parent solutions are influenced by their children through so-called *anterpolation* (Clark and Farley, 1984; Clark and Hall, 1991; Sullivan et al., 1996). The term anterpolation was coined by Sullivan et al. (1996) although the concept is older.

The child-to-parent anterpolation can be implemented using, for instance, the post insertion (PI) approach (Clark and Hall, 1991) where the parent solution is replaced by the child solution within the volume occupied by both domains. In practice, some

buffer zones where anterpolation is omitted are necessary near the child boundaries to avoid growth of unphysical perturbations near the child boundaries (Moeng et al., 2007). An example of a two-way coupled nesting implemented in the WRF-LES model is given by Moeng et al. (2007), and later successfully applied to a stratocumulus study by Zhu et al. (2010). The WRF-LES nesting system can also be used in one-way coupled mode (Mirocha et al., 2013), and this way it has been applied, e.g. to a complex-terrain study (e.g. Nunalee et al., 2014; Muñoz-Esparza et al., 2017) and to an offshore convective boundary layer study (Muñoz-Esparza et al., 2014). Subsequently, Daniels et al. (2016) introduced a vertical interpolation into the WRF model, but this method is restricted to one-way coupled nesting. Recent implementation of an immersed boundary method has made WRF-LES with one-way coupled nesting approach applicable to obstacle-resolving LES (Wiersema et al., 2020). In addition to WRF-LES, the numerical weather prediction model ICON (Zängl et al., 2015) features an LES mode and includes an online nesting capability (Heinze et al., 2017). However, due to its terrain following coordinate system, ICON-LES cannot resolve sharp obstacle structures.

Recently, Huq et al. (2019) implemented a purely vertical nesting system into PALM in which the child and parent domains are required to have the same horizontal extent. Although this approach is useful, e.g. when the grid resolution near the surface needs to be refined to better capture the atmosphere-surface exchange, the requirement of equal horizontal domain extensions poses high demands on the computational resources, limiting this approach to only academic studies. This implementation is also limited to have a single child domain only. For these reasons, we decided to develop the present, more general and fully three-dimensional nesting system in PALM. It can also be run in a pure vertical nesting mode.

One-way coupled obstacle-resolving LES has been applied to a built environment by Nakayama et al. (2016) and by Vonlanthen et al. (2016, 2017). Also the present PALM implementation has already been demonstrated by Maronga et al. (2019, 2020) and applied to obstacle-resolving urban studies (Kurppa et al., 2019; Auvinen et al., 2020a; Karttunen et al., 2020; Kurppa et al., 2020) using the one-way coupling. At current stage, we are not aware of any research on obstacle-resolving LES in the ABL context employing two-way coupled nesting approach. Through our studies, we have observed that the application of two-way coupling in obstacle-resolving LES can become problematic. Therefore, in addition to documenting and evaluating the newly implemented nesting method in the PALM model, this paper addresses the applicability of the two-way coupled nesting approach in obstacle-resolving LES.

The paper is organized as follows: Section 2 gives a brief description of the LES mode of the PALM model system 6.0. Section 3 presents the technical, algorithmic and numerical aspects of the implemented nesting. In Sect. 4 the implemented nesting is evaluated for a series of test cases featuring different kinds of boundary-layer flow. Finally, Sect. 5 summarizes the results and gives and outlook of future developments.

## 2 The PALM model system 6.0 (LES mode)

The PALM model system (Raasch and Schröter, 2001; Maronga et al., 2015, 2020) is based on the non-hydrostatic, filtered, Navier-Stokes-equations in the Boussinesq-approximated or anelastic form. It solves the prognostic equations for the conservation of momentum, mass, energy, and moisture on a staggered Cartesian Arakawa-C grid. Subgrid-scale turbulence is

parameterized using a 1.5-order closure after Deardorff (1980) in the formulation of Saiki et al. (2000). In its standard configuration PALM has thus seven prognostic quantities: the velocity components $u_i$ (where $u_1 = u, u_2 = v, u_3 = w$), the potential temperature $\theta$, specific humidity $q_v$, a passive scalar $s$, and the subgrid-scale (SGS) turbulent kinetic energy $e$. By default, discretization in time and space is achieved using a third-order Runge-Kutta scheme after Williamson (1980) and a fifth-order advection scheme after Wicker and Skamarock (2002). The horizontal grid spacing is always equidistant, whereas it is possible to use variable grid spacing in the vertical direction. Often, the vertical grid spacing is set equidistant within the boundary layer, and stretching is applied above the boundary layer to save computational time in the non-turbulent free atmosphere. At the lateral boundaries cyclic conditions or more advanced in- and outflow conditions can be employed.

Both the Boussinesq and the anelastic approximation require incompressibility of the flow. To provide this feature a predictor-corrector method is used where an equation is solved for the modified perturbation pressure after every Runge-Kutta sub-time step (e.g. Patrinos and Kistler, 1977). The method involves the calculation of a preliminary prognostic velocity. Divergences in the flow field are then attributed solely to the pressure term, leading to a Poisson equation for the perturbation pressure. In case of cyclic lateral boundaries, the Poisson equation is solved by using a direct fast Fourier transform (FFT) method. However, in case of non-cyclic boundary conditions, an iterative multigrid scheme is used (e.g. Hackbusch, 1985).

Parallelization of PALM is achieved by using the Message Passing Interface (MPI, e.g. Gropp et al., 1999) and a two-dimensional (horizontal) domain decomposition.

PALM offers several embedded models to simulate physical processes within the urban environment. Namely, without intention to provide an exhaustive list, this embraces a land-surface (Gehrke et al., 2020) and a building surface model (Resler et al., 2017) to consider the surface-atmosphere exchange of heat and moisture; a radiative-transfer model (Krč et al., 2020) to include complex three-dimensional mutual radiative interactions between surfaces and plants; an indoor and building-energy demand model; an aerosol (Kurppa et al., 2019) and an air-chemistry model (Khan et al., 2020); as well as an embedded Lagrangian particle model for dispersion. For a complete overview we refer to Maronga et al. (2020).

## 3 Nesting system

### 3.1 General concept

The nesting system we have developed is based on the concept of parent and child domains. Each parent domain can enfold multiple child domains but a child domain can, naturally, only have one parent domain. The top-level domain, also called root domain, acts as a parent domain to child domains at the first nesting level. The child domains at first nesting level might have subsequent child domains for which they then act as parent domains (cascading arrangement), see Fig. 1. Our nesting system allows for up to 63 nested domains plus the root domain. The implementation requires that all child domains are always completely located inside their respective parent domain. Also, the grid spacings of a child domain naturally have to be smaller than the grid spacings of its parent domain. The grid-spacing ratios $\Delta X_i/\Delta x_i$ must always be integer valued although different ratios may be used in different directions. Therefore in nested runs the grid stretching is only allowed in the root domain and only above the top boundary of the highest nested domain. There may be multiple child domains at the same nesting levels, but

overlapping child domains at the same nesting level are not permitted. Finally, all the nest domains have to be surface-bound so that elevated child domains are not allowed. Time synchronization is taken care by simply selecting the minimum of the

time steps determined by each model independently and broadcasting this time-step value for all models. Each model inputs and outputs in the same way.

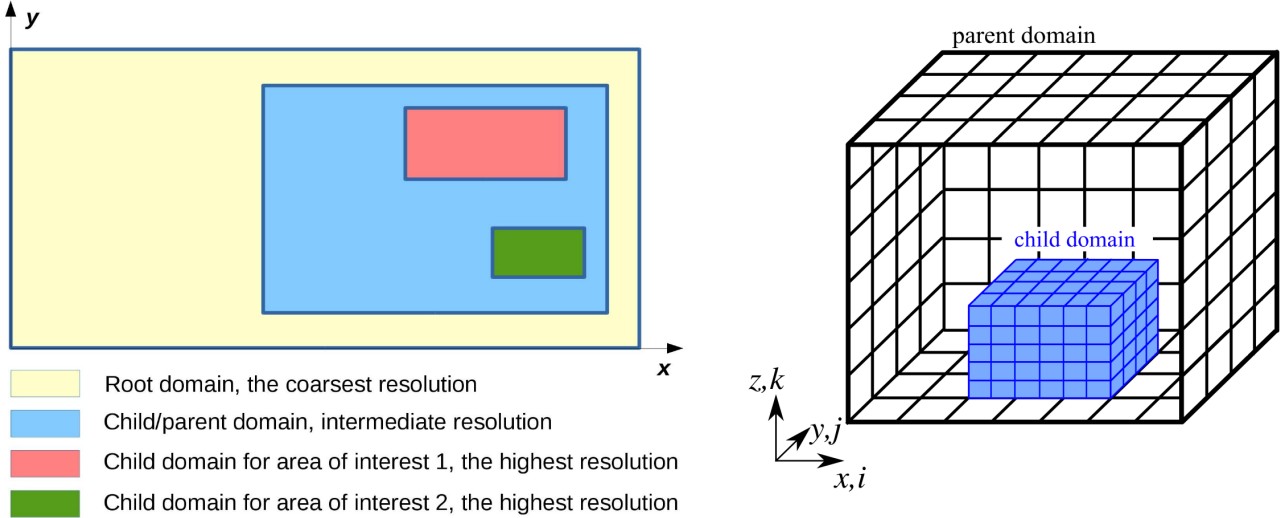

**Figure 1.** A schematic example of a nested configuration involving both cascading and parallel child domains is shown on $x, y$-plane on the left-hand side. On the right-hand side, a three-dimensional view of a nested child domain inside its parent domain is shown.

    In general, the system is designed as two-way coupled nesting, in which a child domain can affect its parent domain and vice versa. It is possible, however, to run the system in a one-way coupled mode where no feedback from the child domain is incorporated in its parent domain. Moreover, it is possible to use the system as a pure vertical one-dimensional nesting, where

the lowest part of the model (e.g. the atmospheric surface layer where the dominant turbulent eddies are usually very small) can be run as a child domain with finer grid spacing than its parent domain that compasses the entire boundary layer. In the case of pure vertical nesting, cyclic boundary conditions must be set on all the lateral boundaries. Unlike the method proposed by Huq et al. (2019), the present method allows a cascade of more than one child domain also in the pure vertical nesting cases.

    The present nesting approach is a variant of the PI method, in which the communication between each parent-child couple

is realized via interpolations (from parent to child) and anterpolations (from child to parent) after each Runge-Kutta sub-step and just before the pressure solver. The latter then ensures that mass conservation is enforced in the anterpolated solution in the parent domain.

## 3.2 Restrictions

The current implementation poses a few restrictions for the nested setups. Moreover, the interpolation and anterpolation methods, which are discussed in the following sections, are based on certain assumptions, e.g. on the grid-line matching between parent and child domains leading to a few more restrictions. Altogether these restrictions are:

– the child domain must always be completely inside its parent domain and there must be a margin of four parent-grid cells between the boundaries of child and parent domains

– parallel child domains must not overlap each other and there must be a margin of at least four child-grid cells between two parallel child domains

– the domain decomposition of all child domains must be such that the sub-domain size is never smaller than the parent grid-spacing in the respective direction

– buildings or any other topography or geometry must not reach the child domain top

– all the grid-spacing ratios must be integer valued

– the outer boundaries of child domains must match with grid planes in its parent domain

– no grid stretching is allowed in the child domains and in root domain it is allowed only above the top boundary of the highest child domain

## 3.3 Structure of the nesting algorithm

Ideally, the coupling actions, i.e data transfers between the domains, anterpolation and interpolation, would be performed after the pressure-correction step using the final divergence free velocity field on both parent and child. To achieve this in the context of the pressure-correction method employed in PALM requires a staged arrangement of the coupling actions such that a child first sends data to its parent and after receiving the data the parent anterpolates and performs the pressure correction step. After the pressure-correction step the parent sends data to the child which interpolates new boundary conditions from the received data and performs the pressure-correction step. The purely vertical nesting method implemented in PALM by Huq et al. (2019) features this kind of staged structure. However, Huq et al.'s method may lead to excessive waiting times as the child has to wait until the parent performs the pressure-correction step and vice versa. Moreover, the staged coupling approach becomes more complicated and more inefficient when a cascade of several nested domains is used. Therefore, Huq's implementation allows for only one child domain. The possibility to employ cascades of child domains was an initial requirement for the present system design and therefore the staged coupling arrangement had to be abandoned. In principle, it would be possible to perform the pressure-correction step twice, first time before the coupling actions for all domains and second time for all parent domains after the coupling to make the anterpolated fields divergence free. However, this would be computationally very expensive severely compromising the benefits from the nesting. This is because the pressure solution is typically the most

time consuming part of the solution process. To avoid this extra penalty, the coupling is based on the preliminary prognostic velocity fields $\boldsymbol{u}_{\mathrm{pre}}$ in the present implementation. The sequence of the coupling actions is illustrated in Fig. 2. This choice has the consequence that the interpolated velocity boundary conditions for a child domain may violate the global mass balance over the child domain such that

$$\int\limits_{S} \rho \widetilde{\boldsymbol{u}}_{\mathrm{pre}} \cdot \boldsymbol{n}\, \mathrm{d}S \neq 0, \tag{1}$$

where the tilde symbol is the interpolation operator and $\boldsymbol{n}$ is the unit inner surface normal vector of the child domain boundary $S$ excluding the bottom boundary. This mass-conservation error, though typically small, is eliminated in an integral sense by adding a constant velocity correction $\Delta\boldsymbol{u}_{\mathrm{pre}}^{(l)}$ on each boundary $l \in \{\mathrm{left, right, south, north, top}\}$

$$\Delta\boldsymbol{u}_{\mathrm{pre}}^{(l)} = -\boldsymbol{n}^{(l)} \frac{\int_{S} \rho \widetilde{\boldsymbol{u}}_{\mathrm{pre}} \cdot \boldsymbol{n}\, \mathrm{d}S}{\int_{S} \rho\, \mathrm{d}S} \tag{2}$$

to the interpolated child boundary values to exactly eliminate the global mass-balance error in Eq. (1). In case of purely vertical nesting mode, the correction is applied only on the top boundary and $S$ spans only over it. This correction is made for all child domains right before the pressure-correction step. According to our tests, $\Delta\boldsymbol{u}_{\mathrm{pre}}$ is typically three or four orders of magnitude smaller than the dominant velocity scales of the flow.

Huq et al. (2019) showed results for a zero mean-wind CBL case. In this case, especially if the nest-top boundary is set on a relatively low level, unphysical overestimation of horizontal velocity component variances easily develop if the coupling is based on $\boldsymbol{u}_{\mathrm{pre}}$. Huq et al. (2019) showed that using the staged sequence of coupling actions, allowing the coupling based on the final velocity field $\boldsymbol{u}$, mostly removes the overestimation of the horizontal-velocity variances. We have confirmed this by temporarily modifying the current implementation to adhere to Huq et al.'s staged arrangement and simulating a vertically nested zero mean wind CBL case similar to Huq et al.'s test case.

In the present method, the overestimation of the horizontal velocity component variances can be mostly avoided by using the integral mass-balance forcing (Eq. 2) and further by setting a narrow buffer zone below the top boundary in which the anterpolation is not performed. This is described in more detail in Sect. 3.5.

In addition to the velocity field, also all other prognostic variables are coupled except the SGS turbulent kinetic energy $e$, as it depends on the resolution by definition and therefore it is not straightforward to couple between parent and child domains having different resolutions. The anterpolated values should be increased by some unknown resolution dependent factor and the interpolated values should be reduced accordingly. $e$ strongly follows the velocity-gradient field and therefore it tends to adapt to the anterpolated velocity field on the parent side during the next Runge-Kutta step without being anterpolated itself. Relying on this reasoning, we omit the anterpolation of $e$. Moreover, we assume that the local generation of $e$ often dominates its advection implying that replacing the interpolation of its child-boundary values by simple zero-gradient conditions may be acceptable. In our numerical tests we compared the zero-gradient conditions with interpolated boundary values reduced by an estimated resolution-difference dependent factor. The comparisons revealed only negligible differences in the results.

Further technical implementation issues are discussed in Appendix A.

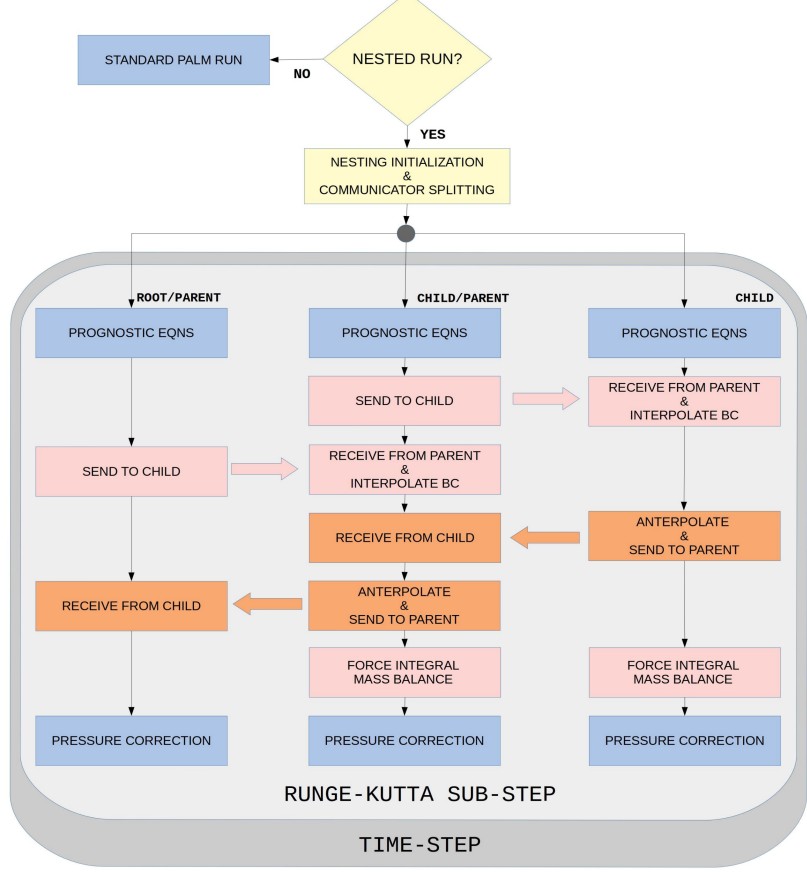

**Figure 2.** Flowchart illustrating the nesting actions in case of three domains in cascading order. In case of more than three levels of domains, more branches similar to the current middle branch would be added. Blue boxes represent baseline PALM actions while the other colors indicate nesting-specific actions. In one-way coupling only the actions indicated by pink color are invoked.

## 3.4 Interpolation (parent to child)

### 3.4.1 Emphasis on conservation properties

Conservation of fluxes through the nest boundaries is an essential condition for a nesting algorithm. Here, by flux conservation we mean that the total flux through a nest-domain boundary is equal to the total flux through the corresponding plane in the parent domain. This must not be confused with the mass-conservation error discussed in Sect. 3.3 Eq. (1), which results from the fact that the nest boundary conditions are interpolated from the preliminary velocity field instead of the final divergence-free velocity field.

Earlier studies by Kurihara et al. (1979) and Clark and Farley (1984) focused mostly on conservation of prognostic variables but not on conservation of their advection fluxes. Later, Sullivan et al. (1996) and Zhou et al. (2018) have paid attention also to

conservation of fluxes, which is very important according to our observations. For example if no attempt is made to minimize the flux-conservation errors of the velocity components on the nested boundaries, the mean-flow angle across the whole system of domains may become significantly deflected. We observed this unacceptable phenomenon in early phase of this work while experimenting with an interpolation scheme which produces non-negligible flux-conservation errors. Therefore, we construct the interpolation method such that the flux conservation errors on the nested boundaries are minimized. In our view, the conservation properties are far more important than local accuracy at the nest boundaries. It will be shown in Sect. 3.4.2 that zeroth-order interpolations are favourable in terms of conservation properties although their local accuracy is obviously lower than those of higher-order interpolations. Here, it should be noted that increasing the order of interpolation accuracy on the child boundary planes is irrelevant because, in all cases, the solution requires a development zone (i.e. a border margin) as it adapts to the increased resolution. Therefore, a low-order interpolation method with acceptable conservation properties should be preferred.

In principle, the most straightforward way to satisfy flux conservation is to directly use the flux on the parent-grid cell face on the nested boundary and to distribute it onto the underlying child-grid cell faces akin to the finite-volume method. However, PALM is based on the finite-difference method and thereby its architecture does not support this method. Therefore, it is necessary to construct an interpolation procedure which is at least approximately flux conservative.

### 3.4.2 General conservation considerations

Before laying out the new interpolation procedure in Sect. 3.4.3, the earlier developments are reviewed and their merits and weaknesses are discussed in this section.

We first consider the work by Kurihara et al. (1979) who required conservation of prognostic variables in the form

$$\Phi_I \Delta S_I = \sum (\phi_i \Delta s_i) \quad \text{for all } i \text{ within the parent-grid cell } I \tag{3}$$

where $\Delta S_I$ and $\Delta s_i$ are the face areas of the parent- and child-grid cells on the nested boundary, and $\sum \Delta s_i = \Delta S_I$. Here the child variables and indices are denoted by lowercase letters and those of parent by uppercase. Later Clark and Farley (1984) applied this condition to both interpolation and anterpolation and called it the reversibility condition. The reversibility condition can also be written as

$$\widehat{\widetilde{\phi(\Phi)}} = \Phi, \tag{4}$$

where the tilde is the interpolation operator and the hat is the anterpolation operator. Based on this condition, Clark and Farley derived a quadratic interpolation scheme that forms a reversible pair with their anterpolation scheme. This reversibility guarantees the *mass*-flux conservation (in incompressible flows). However, it does not guarantee conservation of *other* fluxes consisting of products of the advected variable and advective velocity component. As recently noted by Zhou et al. (2018), the conservation of other fluxes is violated if both advective velocity component and advected variable are interpolated using the Clark and Farley scheme. (Although not mentioned by Zhou et al., this actually applies to any interpolation scheme of higher than zeroth order due to the non-linearity of the advection terms.) Instead of applying the reversibility requirement, Zhou et al.

require global conservation of both prognostic variable $\phi$ and its resolved-scale turbulent flux $\langle u^{(\mathrm{N})\prime}\phi^\prime \rangle$ through the boundary as

$$\langle \phi \rangle_b = \langle \Phi \rangle_b \,, \tag{5}$$

$$\langle u^{(\mathrm{N})\prime}\phi^\prime \rangle_b = \langle U^{(\mathrm{N})\prime}\Phi^\prime \rangle_b \,, \tag{6}$$

where the superscript (N) refers to a boundary-normal velocity component, and $\langle \cdot \rangle_b$ denote spatial averaging over the child domain boundary. It is straightforward to show that if Eq. (5) holds, the flux conservation condition Eq. (6) can be also be written for the total flux as

$$\langle u^{(\mathrm{N})}\phi \rangle_b = \langle U^{(\mathrm{N})}\Phi \rangle_b \,. \tag{7}$$

We shall study the flux conservation using this condition instead of Eqs. (5) and (6).

In order to ensure flux conservation for all prognostic variables, Zhou et al. chose to use the Clark and Farley method only for the advective velocity component and for all advected variables the simple zeroth-order method

$$\phi_i = \Phi_I \quad \text{for all } i \text{ within the parent-grid cell } I \,. \tag{8}$$

Obviously this zeroth-order interpolation satisfies also the reversibility condition Eq. (4) in addition to Eq. (5). This choice readily satisfies the flux conservation condition for all variables. However, in cases with even-valued grid-spacing ratios, Eq. (8) cannot be applied to the velocity components which are in staggered positions relative to $u^{(\mathrm{N})}$ and $U^{(\mathrm{N})}$ on the boundary plane. These velocity components are denoted by $u^{(\mathrm{S})}$ and $U^{(\mathrm{S})}$. Equation (8) is not applicable in this case because the odd number of child-grid $u^{(\mathrm{S})}$-nodes cannot be evenly associated to the surrounding $U^{(\mathrm{S})}$-nodes, see Fig. 3. The method by Zhou et al. (2018) is, indeed, strictly limited to odd-valued grid-spacing ratios.

### 3.4.3 Construction of an approximately flux-conservative method

In our view, the limitation of the method by Zhou et al. is too restrictive. However, we were not able to find any alternative zeroth-order interpolation that would *exactly* satisfy the condition Eq. (7) also for $u^{(\mathrm{S})}$ and be applicable to even-valued grid-spacing ratios. Instead, we found a zeroth-order interpolation which approximately satisfies the condition Eq. (7) for $u^{(\mathrm{S})}$ as will be shown below. This interpolation reads as follows

$$u_i^{(\mathrm{S})} = \begin{cases} U_I^{(\mathrm{S})} & \text{for grid nodes } i \text{ co-located with a parent-grid node } I \text{ in the direction of the interpolation} \\ \frac{1}{2}(U_{I-1}^{(\mathrm{S})} + U_I^{(\mathrm{S})}) & \text{for all grid nodes } i \text{ between parent-grid nodes } I-1 \text{ and } I \,. \end{cases} \tag{9}$$

This interpolation is applicable for $u^{(\mathrm{S})}$ also in cases of even-valued grid-spacing ratios since the child-grid nodes between the parent-grid nodes need not to be associated to the surrounding parent-grid nodes, see Fig. 3.

As another difference to the Zhou et al. method, we do not employ the quadratic scheme of Clark and Farley at all. The reason is that in PALM the interpolation algorithm has to cope with complex geometries, and the application of such more complicated wide-stencil interpolation scheme might become excessively cumbersome. Instead, we use Eq. (8) for the advective velocity

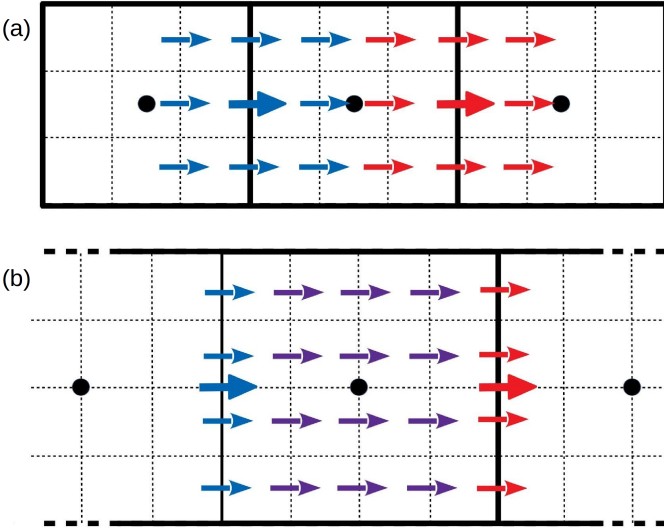

**Figure 3.** Staggered velocity-component nodes in cases of odd (3) a) and even (4) b) grid-spacing ratios. The staggered velocity-component nodes are shown as arrows, thick arrows are for the parent grid and thin ones for the child grid. The parent scalar grid-cell faces are drawn with solid lines and the corresponding child-grid cells with dotted lines. Locations of the corresponding parent-grid scalar nodes are shown as black dots. The blue color indicates the left-hand parent-grid node and red the right-hand node. In case a) the child-grid values are obtained from (8) which cannot be applied in case b). The violet colored child grid nodes in b) receive the averaged values according to Eq. (9).

component $u^{(\mathrm{N})}$, and Eq. (9) for the advected components $u^{(\mathrm{S})}$ and Eq. (8) for all other advected variables. Equation (9) is used for the staggered velocity components also in cases of odd-valued grid-spacing ratio even though Eq. (8) would also be applicable in such cases.

As stated above, the flux conservation condition is satisfied approximately for $u^{(\mathrm{S})}$ for both odd- and even-valued grid-spacing ratios by using Eq. (9) for $u^{(\mathrm{S})}$ and Eq. (8) for $u^{(\mathrm{N})}$. As an example and for the sake of clarity but without any loss of generality, we demonstrate the spatially averaged fluxes of $v$- and $V$-components for a nested domain boundary and assume that the boundary-normal direction is $x$. The velocity components in the $x$-direction are $u$ and $U$. Within the advection scheme, the advective $u$ velocity for the flux is interpolated linearly to the flux point of $v$ as $(u_{j-1}+u_j)/2$. This becomes simply $U_{J-1}$ for those child flux points of $v$ which are located between $V_{J-1}$ and $V_J$, and $(U_{J-1}+U_J)/2$ for those $v$-flux points coinciding with $V_J$. Let us begin by applying the chosen interpolation technique to an extremely simplified example with only two $U$-grid cells along the left nest boundary. Then the $y$-averaged flux $\langle uv \rangle_y$ (we omit the $z$-averaging at this stage) consists of only one $V$-node $J$ in the parent grid. Let the integer-valued grid-spacing ratio in the $y$-direction be $R_y = 4$. We also omit the SGS-fluxes which are assumed small as in the entire discussion above. With these choices $\langle uv \rangle_y$ consists of seven child-grid nodes as

$$\langle uv \rangle_y = \frac{1}{7} \left( 3U_{J-1} \frac{V_{J-1}+V_J}{2} + \frac{U_{J-1}+U_J}{2} V_J + 3U_J \frac{V_J+V_{J+1}}{2} \right). \tag{10}$$

The first term represents fluxes from those child flux points lying between $V_{J-1}$ and $V_J$, the second term is the flux at the child flux point coinciding with $V_J$ and the last term is a similar contribution as the first term but from the child flux points lying between $V_J$ and $V_{J+1}$. Equation (10) can be rearranged as

$$\langle uv \rangle_y = \frac{1}{7} \left[ 4 \frac{U_{J-1} + U_J}{2} V_J + 3 \left( \frac{U_{J-1}}{2} V_{J-1} + \frac{U_J}{2} V_{J+1} \right) \right], \tag{11}$$

where we can identify the factor 4 in the inner term as $R_y$ and the factor 3 in the edge terms as $R_y - 1$. This can be generalized to arbitrary integer $R_y$ and to arbitrary number of parent grid nodes across the boundary $N_y = J_n - J_s + 1$. By doing so, and by incorporating also the $z$-averaging, we obtain

$$\langle uv \rangle_b = \frac{1}{[R_y(N_y+1)-1]N_z} \left[ R_y \sum_{J=J_s}^{J_n} \sum_{K=K_b}^{K_t} \frac{U_{J-1,K} + U_{J,K}}{2} V_{J,K} + (R_y-1) \sum_{K=K_b}^{K_t} \left( \frac{U_{J_s-1,K}}{2} V_{J_s-1,K} + \frac{U_{J_n,K}}{2} V_{J_n+1,K} \right) \right]. \tag{12}$$

On the parent grid, the correspondingly averaged advection flux of $V$ is (as expanded by $R_y$ for easier comparison)

$$\langle UV \rangle_b = \frac{1}{R_y N_y N_z} R_y \sum_{J=J_s}^{J_n} \sum_{K=K_b}^{K_t} \frac{U_{J-1,K} + U_{J,K}}{2} V_{J,K}. \tag{13}$$

Here, $N_z = K_t - K_b + 1$ and $(K_b, \ldots, K_t)$ is the vertical parent-grid index range covering the nest boundary. Clearly Eqs. (12) and (13) are not exactly equal because of the additional edge terms in Eq. (12) containing $V_{J_s}$ and $V_{J_n+1}$, and because the denominator of Eq. (12) deviates from $R_y N_y N_z$. It is important to note, however, that $\langle uv \rangle_b - \langle UV \rangle_b$ tends towards zero as $N_y$ becomes large. In typical applications, the order of magnitude of $N_y$ is hundreds making the flux conservation error negligibly small. Moreover, if we can assume that

$$\frac{1}{N_z} \sum_{K=K_b}^{K_t} \left( \frac{U_{J_s-1,K}}{2} V_{J_s-1,K} + \frac{U_{J_n,K}}{2} V_{J_n+1,K} \right) \approx \frac{1}{N_y} \sum_{J=J_s}^{J_n} \frac{1}{N_z} \sum_{K=K_b}^{K_t} \frac{U_{J-1,K} + U_{J,K}}{2} V_{J,K} \tag{14}$$

which is a reasonable assumption in many cases, then $\langle uv \rangle_b - \langle UV \rangle_b \approx 0$ even with small values of $N_y$.

### 3.4.4 Effects of the advection scheme on flux conservation

Above, the flux conservation was discussed generally without taking into account the effects of the actual discretization scheme emplyed in the advection algorithm. In this subsection, a mismatch of the advection-term approximations on the child and parent sides in PALM is first identified and discussed, and subsequently a method to reduce the flux-conservation error resulting from this mismatch is proposed.

In Sect. 3.4.3 it was assumed that the advected parent-grid variable values on the child boundary ($V_{J,K}$ in the above example Eqs. (12) and (13)) are equal to the values used for the fluxes in the advection-term computation. This is not the case in PALM as they are interpolated onto their flux points using the fifth-order scheme by Wicker and Skamarock (2002). Here, it is important to understand, that the interpolation onto the flux point in the advection scheme is a separate procedure from the

parent-to-child interpolation, and it is performed in a different phase of the time-step cycle (Fig. 2) in the prognostic equations step. In PALM, the fifth-order interpolation is not employed at the boundaries, except cyclic boundaries, instead so-called advection-scheme degradation procedure is utilized. The degradation procedure entails degrading the flux-point interpolation within the advection scheme on first two layers of nodes. The first-order upwind scheme is applied on the first layer and the third-order Wicker and Skamarock (2002) scheme for the second layer. This way, only one layer of boundary ghost points is needed at the boundary. Technically, three grid layers are allocated in PALM for boundary ghost points, but using all of them at child boundaries would lead to no gain in accuracy as the second and third layers would be just copies of the first layer due to the zeroth-order parent-to-child interpolation.

The first-order upwind scheme makes the advected values on child-boundary flux points independent of the child solution itself, if the local flow direction is into the child domain. This is important from the flux-conservation point of view as the flux into the child domain should be entirely controlled by the parent solution. On the other hand, the first-order upwind scheme leads to values on the child-boundary flux points that may differ from those on the corresponding grid plane on the parent side as those are interpolated with the fifth-order scheme. Therefore, additional flux-conservation errors may be generated.

We have not found any way to totally eliminate the resulting additional flux-conservation error, but we can reduce it in the following way. Instead of using the original parent-grid values in the parent-to-child interpolation, we replace them by values pre-interpolated (Fig. 4, phase 1) onto the parent flux points using a scheme of higher than first order, and use these values in the parent-to-child interpolation. From here on, we refer to this pre-interpolation as transfer to boundary plane (TBP). As a result, the formally first-order upwind advection scheme becomes the selected higher order scheme if the local flow direction is into the child domain.

The TBP must not employ more than one parent-grid layer behind a child-domain boundary because the child has no information about the parent-domain geometry outside the first parent-grid layer. An interpolation stencil reaching further away could penetrate a vertical wall leading to erroneous interpolation. Therefore, the best available choice is to simply use the average of the parent-grid values on both sides of the child-domain boundary, i.e. a second-order interpolation. Obviously it is different from the fifth-order scheme, but we argue that the difference between values interpolated onto the boundary plane using the fifth-order and second-order schemes can be expected to be smaller than the difference between those interpolated using the fifth-order and first-order schemes. Our numerical tests support this argument.

On the top boundary, there is no geometry and hence we can use wider TBP-stencil there. We ended up using the third-order Wicker and Skamarock (2002) scheme there for the TBP because, in our numerical tests it yielded almost indistinguishable results from the more complicated and more communication-intensive fifth-order scheme. Note, that the TBP reduces the flux-conservation error only on those boundary regions where the flux is into the child domain.

The sequence of interpolation operations is illustrated in Fig. 4 using the left child boundary as an example. Phase-1 operations belong to the TBP and phase-2 operations to the actual parent-to-child interpolation using Eqs. (8) and (9). Note that TBP is not applied to the boundary-normal velocity component $u^{(\mathrm{N})}$.

We evaluated the flux-conservation error in a simple test run modelling a horizontally homogeneous slightly convective boundary layer over flat terrain with capping inversion at $z = 450$ m. The constant kinematic surface heat flux is $0.025 \ \mathrm{Kms^{-1}}$

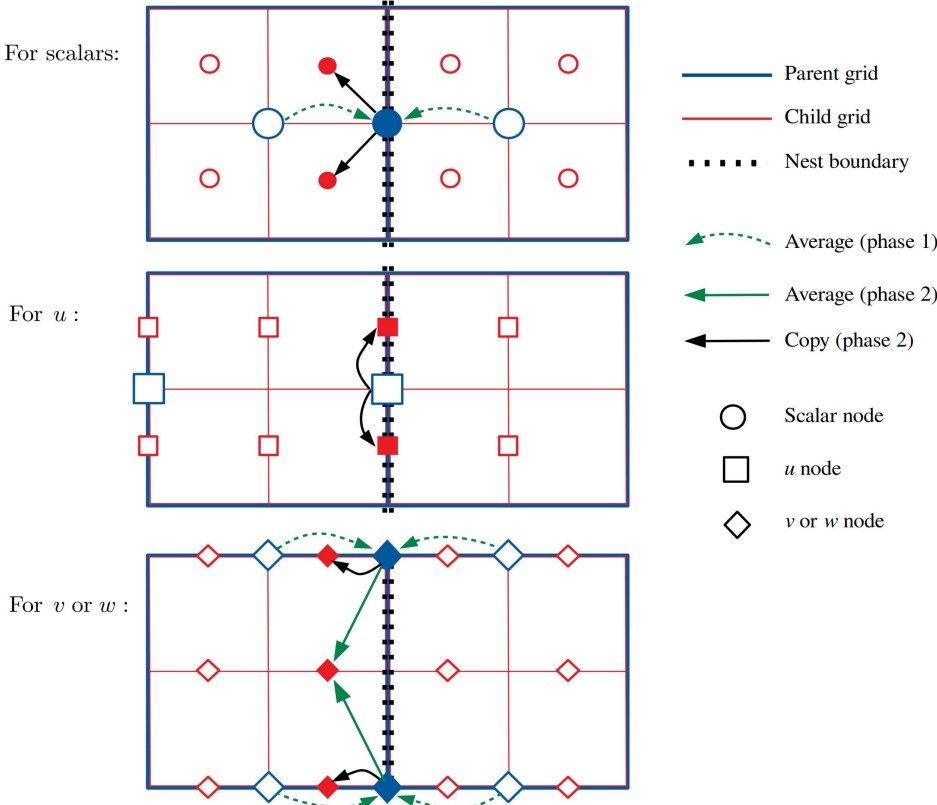

**Figure 4.** A schematic illustration of the interpolation operations on the left child boundary as an example. The child-grid nodes on the left side are boundary ghost nodes. Phase-1 operations belong to the TBP and phase-2 operations to the actual parent-to-child interpolation using Eqs. (8) and (9). Filled blue symbols denote intermediate values on the boundary plane (flux plane) resulting from the TBP. Filled red symbols denote the final boundary-condition values for the child boundary. In this example $u$ is $u^{(N)}$ while $v$ and $w$ are $u^{(S)}$.

and the wind is driven by the geostrophic balance with geostrophic mean wind angle of $11°$ relative to the $x$-axis. Within the boundary layer the mean wind angle is close to $40°$ degrees making the $u$- and $v$-components roughly equal to each other in average. The root grid dimensions are $256 \times 256 \times 48$ in $x$-, $y$- and $z$-directions, respectively, with isotropic grid spacing of 12 m. The nest domain is placed in the middle of the root domain and its grid dimensions are $256 \times 256 \times 64$. The nest grid spacing is isotropic 6 m, thus $R_x = R_y = R_z = 2$. Periodic boundary conditions are given on the lateral root boundaries. The

$u$ and $v$ flux-conservation errors are evaluated for the left nest boundary for which $N_y = 128$ and $N_z = 32$. The relative errors $(\langle uu \rangle - \langle UU \rangle)/\langle UU \rangle$ and $(\langle uv \rangle - \langle UV \rangle)/\langle UV \rangle$ fluctuate in time within $\pm 3$ % with average of 0.04 % and root-mean square value 1.7 % for $(\langle uu \rangle - \langle UU \rangle)/\langle UU \rangle$, and 0.02 % and 0.6 % for $(\langle uv \rangle - \langle UV \rangle)/\langle UV \rangle$, respectively. It is expected that with larger grid dimensions these errors will become even smaller.

According to our numerical tests presented in the Sect. 4, the proposed zeroth-order intepolation method, Eqs. (8) and (9)
together with TBP, has proved fully sufficient and remains the only nesting-interpolation method implemented in PALM. We

have considered also an alternative interpolation approach for the advected variable based on tri-linear interpolation with a specific reversibility correction. Although it is not implemented, a short discussion is provided in Appendix B.

## 3.5 Anterpolation (child to parent)

Anterpolation is used to feed the child domain solution back to its parent domain. Generally, anterpolation consists of filtering the fine-grid child solution $\phi_{i,j,k}$ and mapping it to the parent-domain grid. We select to employ the anterpolation scheme proposed by Clark and Farley (1984), which consists of simple averaging over one parent-domain grid volume around the parent-grid node of the variable in question corresponding to top-hat filtering, viz.

$$\widehat{\phi}_{I,J,K} = \frac{1}{N_{I,J,K}} \sum_{i_1(I)}^{i_2(I)} \sum_{j_1(J)}^{j_2(J)} \sum_{k_1(K)}^{k_2(K)} \phi_{i,j,k} \,. \tag{15}$$

The original parent solution $\Phi_{I,J,K}$ is replaced by the anterpolated solution in the domain of overlap. Here, $i,j,k$ and $I,J,K$ are the child- and parent-grid indices, respectively, and the hat is the anterpolation operator. The summation index limits, i.e. the span of the anterpolation cell $i_1(I)$, $i_2(I)$, $j_1(J)$, $j_2(J)$, $k_1(K)$ and $k_2(K)$ are pre-computed during the initialization and they depend on the grid configuration and the variable in question, i.e. the staggered velocity components have different index limits than the grid-cell centered scalars. Note that for the velocity components, the anterpolation volume is reduced to the grid-cell face on which the velocity component is defined. This means that the upper index limit in the direction of the velocity component is reduced to the lower one, for instance $i_2 = i_1$ for $u$, because the coordinates of the velocity component node in the respective direction in the parent and the child readily coincide, thus there is no need for anterpolation in this direction. $N_{I,J,K}$ is the number of child domain values used for anterpolation at a given parent-grid location, and is pre-computed during the initialization as

$$N_{I,J,K} = [i_2(I) - i_1(I) + 1][j_2(J) - j_1(J) + 1][k_2(K) - k_1(K) + 1] \,. \tag{16}$$

Note that due to the staggered grid, four sets of the index limits and $N_{I,J,K}$ are pre-computed and stored: one for each velocity component and one for all scalars. Generally, the anterpolation cells can be spanned in more than one way. We define the anterpolation cells similarly to Clark and Farley (1984) in order to ensure the reversibility discussed in Sect. 3.4.2. For scalar variables (non-staggered variables) the anterpolation cell spans $X_i \pm \Delta X_i/2$ where $X_i$ $(i = 1,2,3)$ are the coordinates of the scalar node in the parent grid. For the velocity components (staggered variables), for example for $u$, the anterpolation cell spans $X_1$, $X_2 \pm \Delta X_2/2$, $X_3 \pm \Delta X_3/2$, where $X_i$ are the coordinates of the staggered $u$-node in the parent grid.

Buffer zones where the anterpolation is omitted are applied next to the child-domain boundaries except the bottom boundary. The main purpose of the buffer zones is to avoid an unstable feedback loop between the anterpolation and interpolation. The default width of these buffer zones is two prognostic grid nodes. The user may choose a different value for the buffer width, but the minimum allowed width is one parent-grid spacing. This is because the layer of nodes nearest to the child boundary is directly used in the interpolation, and using an anterpolated value for interpolation leads to a strongly unstable behaviour. The buffer zones are comparable to the relaxation zones applied in the nesting system of the WRF-LES model (Moeng et al.,

2007). In the WRF-LES nesting system the anterpolation is under-relaxed within these zones such that the under-relaxation coefficient varies linearly across the relaxation zones which are five grid spacings wide. As mentioned in Sect. 3.3 the buffer zone below the top boundary also reduces the overestimation of the horizontal velocity variances observed in zero mean-wind CBL tests in purely vertical nesting mode. According to these tests in purely vertical nesting mode, simulation results are not particularly sensitive to the extent of the vertical downward shift of the upper edge of the anterpolation domain.

**Canopy-restricted anterpolation**

The anterpolation algorithm is implemented in the PALM model with a feature that enables its application in a spatially selective manner such that the operation is only performed within the computational domain that is above a user-defined vertical threshold. This practice is discovered to resolve complications that arise when two-way coupled nesting is applied in obstacle-resolved LES simulations where the anterpolated solution within the obstacle canopy introduces discrepancies in the coarser parent solution. Thus we label this approach *canopy-restricted* (CR) anterpolation and the coupling is referred to as two-way CR, for short. The necessity of this anterpolation strategy is motivated and its effectiveness demonstrated in Sect. 4.2.3 where nesting is applied to obstacle-resolved LES test case.

# 4   Numerical experiments

In order to evaluate the nesting strategy, to show its benefits and point out its limits, we performed a series of nested model simulations for different grid-spacing ratios and respective non-nested reference simulations for different atmospheric situations. The idea is not to mainly validate the PALM model against experimental data, but to systematically compare the nested-domain results to corresponding non-nested fine- and coarse-grid reference results, and to show that the nested-domain solutions are closer to the fine-grid reference solutions than the coarse-grid reference solutions are. PALM has been already evaluated for various ABL flows against measurement data (Letzel et al., 2008). Nevertheless, we show one comparison against wind-tunnel data in Sect. 4.2.2. Furthermore, the idea is not to present grid convergence studies since the grid convergence of the PALM model has been demonstrated previously e.g. for convective boundary layer by Hellsten and Zilitinkevich (2013).

We simulated a homogeneously-heated flat-terrain convective boundary layer as well as a purely shear-driven flat-terrain boundary layer. Further, to investigate the performance of the grid nesting in more complex situations where non-flat topography is present, we performed two-staged nested simulations for a neutrally-stratified flow over a smooth three-dimensional hill and will compare the results against wind-tunnel data. Second, we simulated a neutrally-stratified urban boundary-layer flow over a regular staggered arrangement of building cubes using one- and two-staged nesting, and will compare the nested simulation results to corresponding non-nested fine- and coarse-grid simulation results. Details concerning the different simulation setups are given in their respective sections. Note that for the sake of simplicity velocity components will hereafter be addressed by lower case variable names only, no matter if it refers to the flow in the parent or the child domain.

## 4.1 Convective boundary layer

The nesting method is first evaluated for a pure convective boundary layer (CBL) with zero mean wind. We set up one child domain that is centered within the parent domain. For the root domain, cyclic lateral boundary conditions were set. A homogeneous and time-constant surface sensible heat flux of $0.1 \ \mathrm{K \, m \, s^{-1}}$ was prescribed. The simulation was initialized with a potential temperature profile that increases linearly with height at a lapse-rate of $0.3 \ \mathrm{K}/100 \ \mathrm{m}$. The root-model domain size is $10.2 \ \mathrm{km} \times 10.2 \ \mathrm{km} \times 3.0 \ \mathrm{km}$ in the $x$-, $y$- and $z$-directions, respectively, with an isotropic grid spacing of $20 \ \mathrm{m}$. The top of the child domain is set to be within the middle part of the CBL, and the domain size is $2.5 \ \mathrm{km} \times 2.5 \ \mathrm{km} \times 0.48 \ \mathrm{km}$ in the $x$-, $y$- and $z$-directions, respectively, with an isotropic grid spacing of $10 \ \mathrm{m}$, resulting in a grid-spacing ratio of 2. In order to examine how turbulence statistics behave for different grid-spacing ratios between parent and child in the CBL, we additionally run nested simulations with grid-spacing ratio of 3 and 4 by increasing the isotropic grid spacing in the parent domain to $30 \ \mathrm{m}$ and $40 \ \mathrm{m}$, respectively. Non-nested coarse- and fine-grid reference simulations were carried out corresponding to the nested simulations with different grid-spacing ratios. The simulated time was four hours for all convective cases. Data analysis started after 2 hours of simulated time when model spin-up effects are not present any more and the simulations reached steady-state conditions. In order to perform a spectral analysis of time-series data, the time step was held constant at $1.0 \ \mathrm{s}$ in all convective simulations during the data analysis period.

Figure 5a) shows an instantaneous horizontal cross-section of the $w$-component at a height of $40 \ \mathrm{m}$ for the parent and child (overlaid) domains for the grid-spacing ratio 2. A hexagonal pattern of convective cells with strong updrafts and weaker downdrafts is visible, as it can be typically observed in LES. The transition between parent and child appears smooth and the flow structures are continuous in terms of shape and amplitude, while within the inner part of the child domain more fine-scale structures can be observed with slightly stronger up- and downdrafts, as also reported by Moeng et al. (2007). Furthermore, Fig. 5b), showing an instantaneous vertical cross-section for the $w$-component, also depicts how the up- and downdrafts are consistently maintained across the child boundary without any obvious impact on the turbulent structures.

Figure 6 shows horizontally- and time-averaged vertical profiles of potential temperature $\theta$, vertical turbulent heat flux $\langle w'\theta' \rangle$, variances of horizontal and vertical velocity components, as well as the skewness of the vertical velocity component $w$, being one of the most grid sensitive quantities (Sullivan and Patton, 2011). The profiles of $\langle \theta \rangle$ indicate a well mixed CBL. With increasing grid-spacing ratio the corresponding parent and coarse grid simulations deviate from the fine grid reference, particularly near the surface and within the inversion layer, while the child results adhere well with the non-nested fine-reference simulation, indicating that the profiles of $\langle \theta \rangle$ in the child domains are rather independent of the parent grid for the employed grid spacings.

The heat flux profiles in the child and parent simulations decrease linearly with height within the CBL and are in good agreement with the fine reference simulation. For the parent simulation we note the near-surface kink in the heat flux (see Fig. 6c for a close-up view). Moeng et al. (2007) observed a similar kink in the heat flux and attributed it to inaccuracies in the statistical evaluation of the heat flux, more precisely, to errors that arise from interpolation from a mass- to a height-coordinate system. However, to evaluate fluxes PALM does not apply any interpolations but uses directly the resolved- and

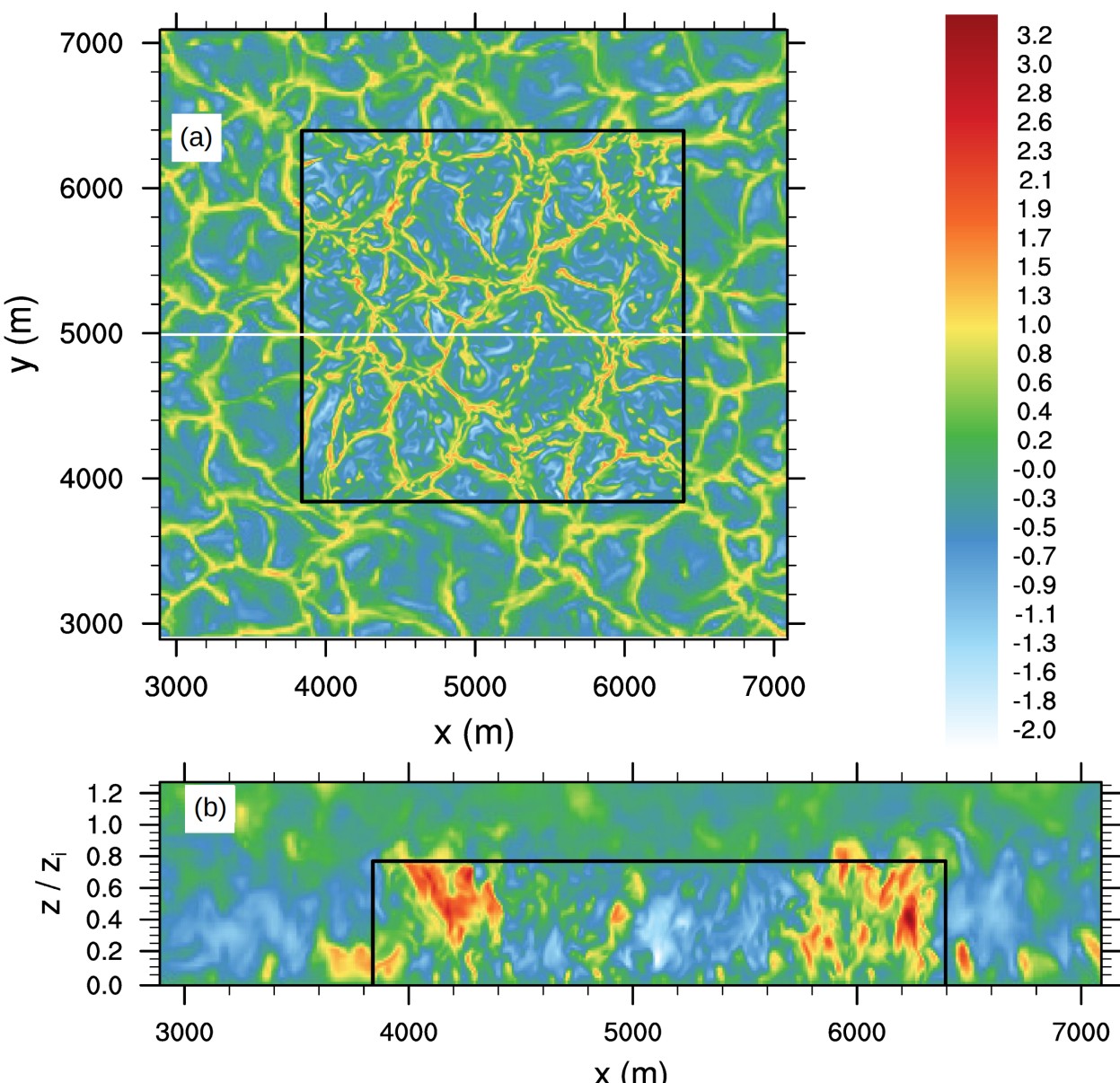

**Figure 5.** Instantaneous horizontal a) and vertical b) cross section of $w$ after 4 h of simulated time for the grid-spacing ratio case 2. The horizontal cross-section is given at a height of 40 m. The black box indicates the lateral and top boundaries of the child domain. The white line indicates the $y$-position of the vertical cross section of $w$ shown in (b). The vertical axis in (b) is normalized with the horizontal mean boundary-layer depth $z_i$. Note that only part of the parent domain is shown for the sake of visibility.

subgrid-scale fluxes as calculated in the advection scheme and the subgrid model, respectively, so that interpolation errors cannot explain the kink in this case. Instead, we attribute the kink in the parent domain to the anterpolation from the fine child

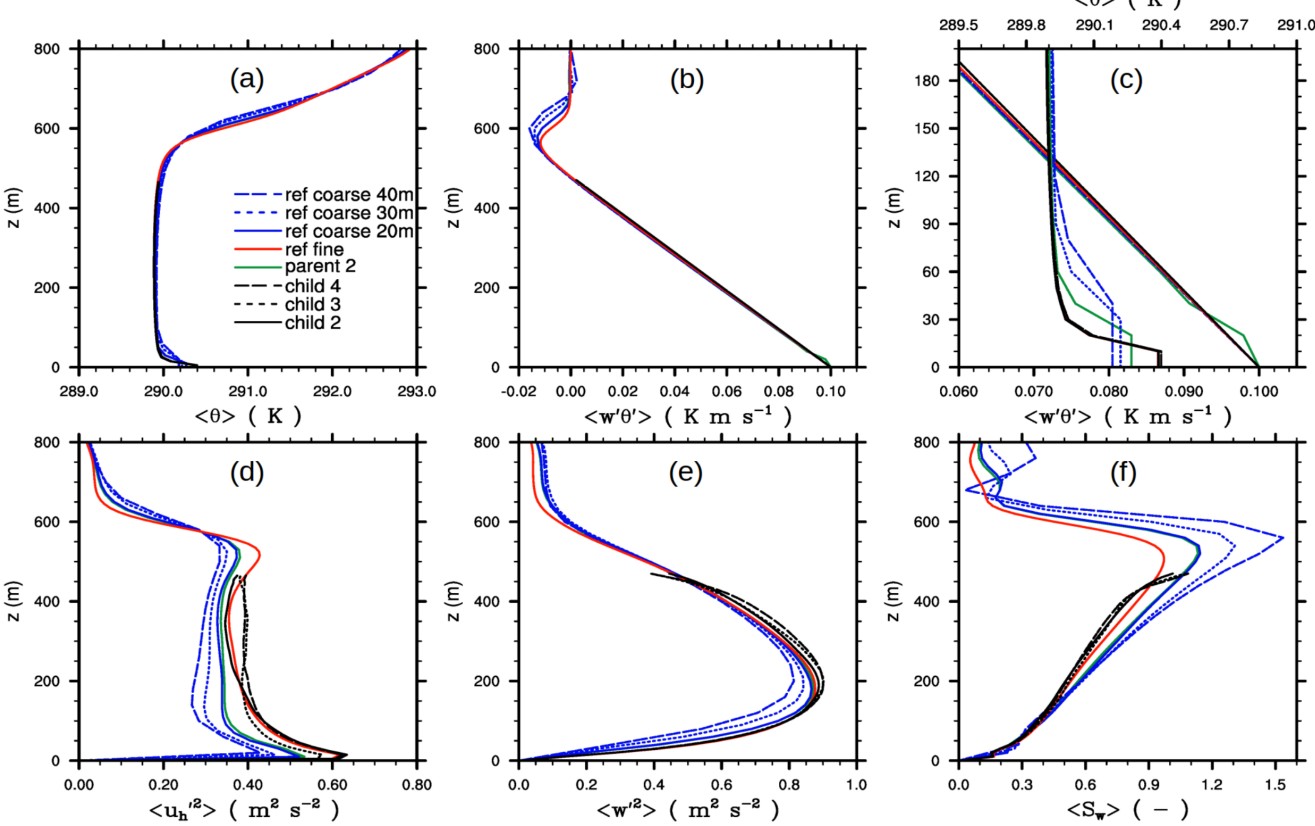

**Figure 6.** 30-min time and horizontally averaged profiles of a) $\langle \theta \rangle$, b) $\langle w'\theta' \rangle$, c) a close-up view of $\langle \theta \rangle$ and $\langle w'\theta' \rangle$, d) variance of the horizontal velocity components, e) variance of the vertical velocity component, and f) skewness of the vertical velocity component after four hours of simulated time. Note the second upper abscissa in c). Profiles are shown for the grid-spacing ratio of 2, 3, and 4 for the respective child domains, indicated by the respective numbers. All these child domains have the same resolution as the fine-grid reference simulation. The corresponding profiles from the coarse-grid reference simulations for the 20 m, 30 m, and 40 m grid spacing are indicated the same. For the sake of clarity, the resulting profiles for the parent domain are only shown for the grid-spacing ratio of 2. Angle brackets indicate a horizontal average over the domain.

solution. In simulations with different vertical grid spacing, the vertical gradients of $\langle \theta \rangle$ within the unstable near-surface layer are differently resolved, resulting in slightly different near-surface temperatures, as it can, e.g., be observed between the fine and coarse reference simulations in Fig. 6 a. This indicates that the parent simulation will yield slightly different $\langle \theta \rangle$-profiles than the child simulation. After the anterpolation is performed, the parent solution is replaced by the underlying child solution, where the near-surface vertical gradients of $\langle \theta \rangle$ in the parent domain partly deviate from the ones the model would create without feedback from the child domain, i.e. the near-surface $\langle \theta \rangle$-profile in the parent is not in equilibrium with the applied surface boundary condition any more. In the following time step the parent model tries to re-adjust the post-inserted $\langle \theta \rangle$ to the vertical gradients as being present without feedback from the child, altering the heating rates and thus the near surface vertical

gradients of the heat flux, which in turn becomes visible as near-surface kink. In fact, we verified this hypothesis in a test case by using identical vertical grid spacing in parent and child. In this case, no kink in the vertical heat flux was visible any more (not shown).

The variances of the horizontal and vertical velocity components, as well the skewness of the vertical component, depend strongly on the grid spacing as the coarse- and fine-grid reference simulations show, where the variances (skewness) become smaller (larger) for increasing grid spacing. The parent simulation agrees well with the coarse-resolution simulation, indicating that the anterpolation changes the parent flow field only marginally. The variances and skewness in the child simulations agree with the fine-reference profiles, except for the upper regions of the child domain where the variances are slightly overestimated. The child profiles are almost independent of grid-spacing ratio, and are close to the reference simulation profile. This indicates that the child solutions are almost independent on the chosen grid-spacing ratio in the studied cases.

Although there is no mean horizontal advection in the zero-mean wind CBLs, spatially and temporally local horizontal advection always takes place and therefore flow structures are advected locally from parent to child (and vice versa). Therefore, advected flow structures may need a certain fetch to adjust to the changed grid spacing. In order to get an idea of how long distance from the lateral child boundaries is required to observe similar turbulence properties as in a non-nested fine resolution reference simulation, we performed a spectral analysis. Therefore, we sampled time series of TKE and $\theta$ at different locations inside the child domain and calculated frequency spectra from the sampled time series. Subsequently, we averaged spectra over all sampling locations with the same distance from the lateral child boundaries. Overall, we calculated spectra inside the child domain at locations 75, 100, 200, 300, and 500 m away from the lateral boundary. It should be noted that transforming frequency spectra into wave number spectra using Taylor's hypothesis in order to directly link spectral information and grid spacing is not strictly correct in this case where we have no background wind; nevertheless we will assume that frequency and wave number space are connected, i.e. large frequencies belong to small spatial scales and vice versa. Figure 7 shows the resulting frequency spectra, as well as corresponding spectra from fine- and coarse-grid reference simulation. As expected, the coarse-resolution spectra exhibit less spectral energy at larger frequencies, compared to the child- and fine-resolution spectra. This is due to the larger filter length assumed for the subgrid model, removing more energy at larger spatial scales and thus also affect smaller frequencies. The child-spectra agree well with the fine-reference spectra, especially for the grid-spacing ratio case 2 where even locations close to the lateral boundaries show good agreement with the reference. We attribute this to the nature of the CBL, where turbulence is mostly produced locally by buoyancy and horizontal advection is almost negligible, so that turbulence is almost not affected by any transport from the boundaries. Also for the grid-spacing ratios of 3 and 4 the differences are generally small, but locations close to the lateral boundaries show slightly smaller spectral densities at larger frequencies, indicating that for larger grid-spacing ratios the flow needs a fetch of a few tens of meters in a purely buoyancy-driven boundary layer to adjust to the finer grid spacing.

Even though the child simulations yield turbulence profiles, spectra and instantaneous flow patterns similar to the fine-grid reference simulation for a pure-buoyancy driven flow, the nested simulation nevertheless creates side effects on the flow which appear as a secondary circulation (SC). This SC is not caused by a violation of mass conservation that has been discussed in Sect. 3.4. Figure 8 shows the 5-hour time-averaged $w$-component at the middle part of the CBL in a homogeneously-

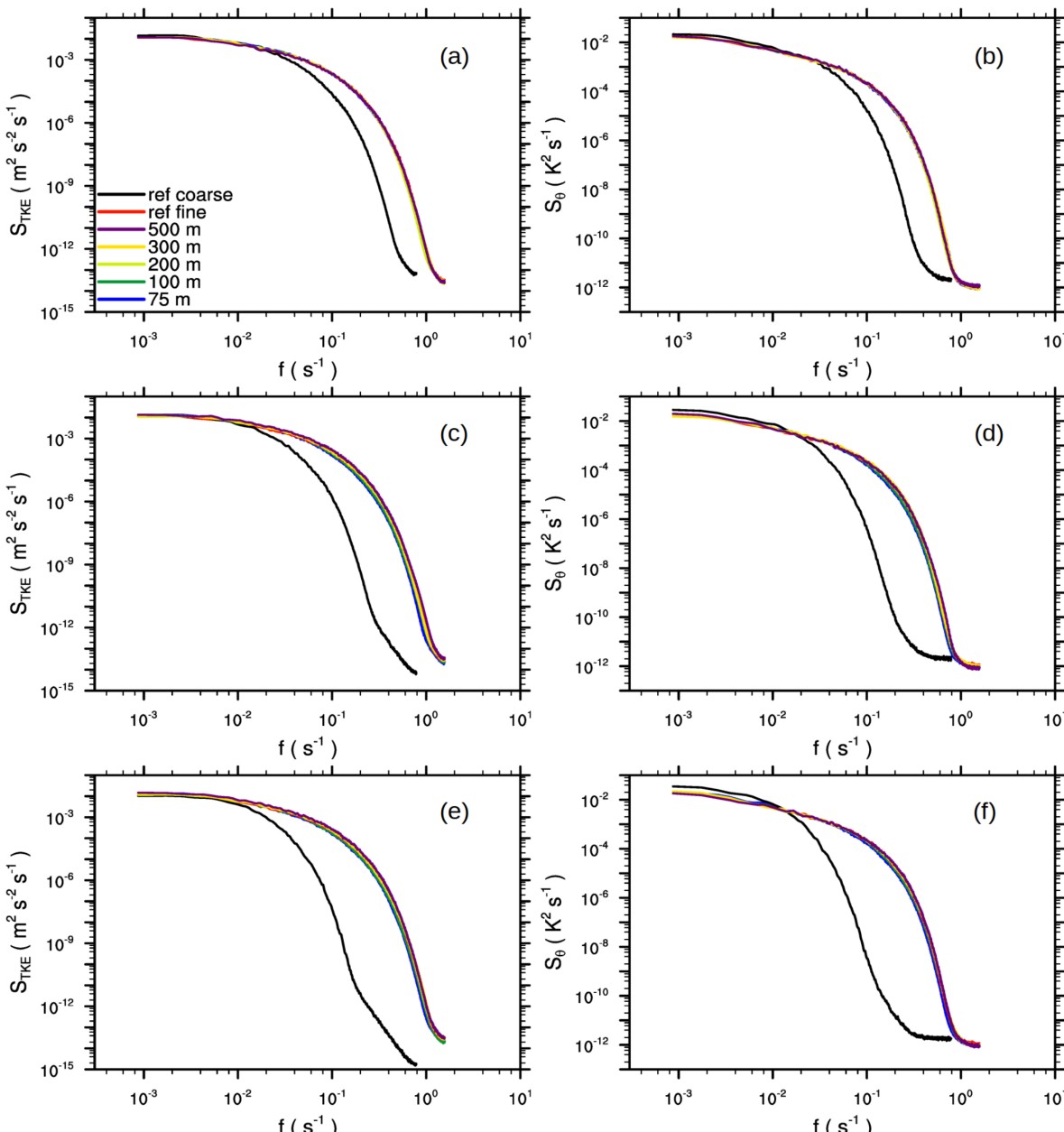

**Figure 7.** Frequency spectra of the TKE (left column) and $\theta$ variance (right column) inside the child domain for the grid-spacing ratio of 2 (a,b), 3 (c,d), and 4 (e,f). The coloured curves represent spectra taken at different distances away from the lateral child boundaries. Furthermore, spectra for fine- and corresponding coarse-grid references simulations are displayed. TKE and $\theta$ were sampled at $z = 120$ m.

heated nested simulation. In order to compute the 5-hour time average, we continued the simulation with grid-spacing ratio 2 for further 3 hours. Within the region of the nested child domain a mean updraft can be observed, which is in the range of

$0.4 - 0.9 \, \mathrm{m\,s^{-1}}$ and extends throughout the entire depth of the CBL (not shown). At the child domain boundaries and outside the child-domain region the flow subsides in average, and horizontally directed branches at the upper and lower parts of the CBL occur, giving the overall picture of a SC. The strength of this SC, indicated by the amplitude of the mean updraft, is in the order of the strength of SCs observed in previous simulations over idealized stripe-like surface heterogeneities (Sühring et al., 2014) and even exceeds the strength of SCs observed in simulations over realistic surface forms (Maronga and Raasch, 2013).

SCs develop above surface heterogeneities mainly due to differential surface heating of the air, resulting in mean updrafts and downdrafts over the stronger- and less-heated patches, respectively. However, since we prescribe the same surface sensible heat flux in the parent as well as in the child simulation, differential surface heating cannot be the reason of the SC in the nested simulation. Moeng et al. (2007) observed a temperature bias in their child domain that led to mean vertical motion to compensate the temperature bias. They observed temperature biases that go either way, i.e. a too cold or a too warm child domain, which they attributed to a nested child domain of too small horizontal extent. If only a few up- or downdrafts are resolved in the child domain, the vertical transport is dominated by these up- or downdrafts and thus a warmer or cooler CBL can be quickly produced in the child domain, respectively. They showed that for larger horizontal child domain size the temperature bias and thus the associated vertical motion vanished. However, they only considered instantaneous differences between parent and child, meaning that the temperature bias is a result of insufficient sampling of the large up- and downdrafts rather than an inherent feature of the nesting which can only been observed after time-averaging. In our case the SC becomes visible only after considerable time-averaging. The updraft branch of the secondary circulations is always located within the child domain also for larger child domain extensions. We hypothesize that this SC is triggered by a slightly different divergence of the vertical heat flux between the region occupied by the child domain and the remaining parent domain due to different grid spacing. It might be impossible to eliminate, because higher resolution better represents the turbulent mixing, so differences between the parent and the child solutions are to be expected in general.

Even though this inherent artificially-induced SC only appears when the flow is averaged over a longer time under quasi-stationary conditions (no diurnal cycle, no change in the mean wind, etc.), nested simulation results should be interpreted carefully in terms of SCs. In particular, since the strength of the artificial SC is in the order of 'real-world' circulations over heterogeneous terrain, these two may become superimposed, altering the pattern of the vertical transport of sensible and latent heat. Although we did not succeed to proof our hypothesis, we encourage other researchers to look for the existence of such SCs in any nested models by analyzing the time averaged results.

## 4.2 Neutrally stratified boundary layer tests

**Initialization and inflow conditions**

As further test cases, we set up a series of boundary layer flow simulations with increasing order of complexity. First, to evaluate the performance of grid nesting in shear-driven boundary-layer flows, we simulated a flow over a homogeneous flat surface in order to compare first and second-order moments from a nested simulation against reference simulations. In a second step, we simulated a flow over a smooth three-dimensional hill for comparison of nested simulation results against wind-tunnel data.

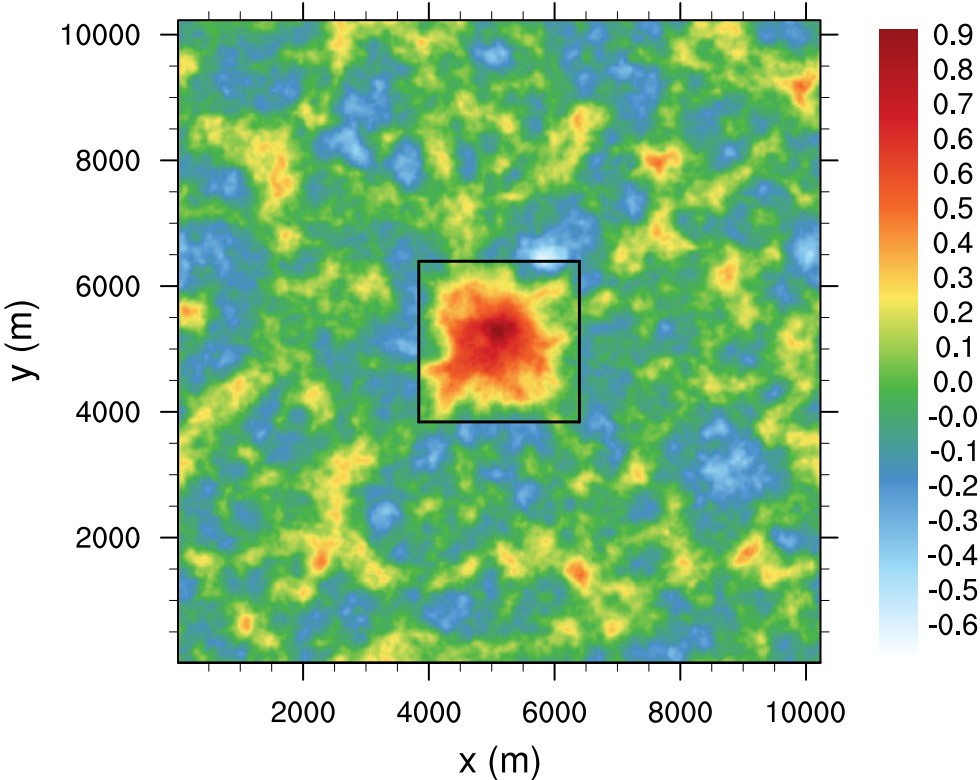

**Figure 8.** Horizontal cross-section of 5-h time-averaged vertical velocity at $z = 400$ m in a nested simulation with grid ratio of 2. The black box indicates the location of the child domain.

Finally, in order to illustrate the advantages of the grid nesting in more complex setups, we simulated a flow over a staggered arrangement of cubes mounted on a flat surface.

The parent domain size for all neutrally-stratified simulations was $L_x \times L_y \times L_z = 5.1 \times 1.5 \times 0.32 \text{ km}^3$ in the $x-, y-,$ and $z$-directions, respectively. In all neutral simulations we prescribed a homogeneous roughness length of $z_0 = 0.01$ m. At the top boundary we applied a free-slip condition for the horizontal wind components and zero vertical motion. At the spanwise lateral boundaries (north and south boundary) we applied cyclic conditions. At the western lateral boundary (hereafter referred to as inflow boundary) we prescribed mean inflow profiles for the $u$ and $v$ component, obtained from a cyclic precursor run. Two different precursor simulations were employed for the subsequent test cases. The one used for the flat surface and for the smooth hill featured a geostrophic wind of $u_g = 4.8 \text{ m s}^{-1}$ and $v_g = -1.3 \text{ m s}^{-1}$ at a latitude of 55 degrees, adjusted such that the surface-layer mean flow became parallel with the $x$-axis. This precursor simulation ran for 36 hours to reach a stationary state. The second precursor simulation, used for the cuboid case, was driven by a fixed pressure gradient angled to result in a mean flow of $u = 10 \text{ m s}^{-1}$ at $z = L_z$ with a 3 degree angle from the $x$-axis.

In order to obtain a turbulent inflow, we applied a turbulence recycling method according to Kataoka and Mizuno (2002), where the inflow mean vertical profiles of $u$ and $v$ are superimposed by turbulent fluctuations sampled at a recycling plane,

which is placed at $x_{rc} = 1.5$ km downstream the inflow boundary. The recycling plane is placed sufficiently far apart from the inflow boundary to allow for statistically-independent turbulence, but also sufficiently far apart from the location of the child domain to avoid any feedback between the grid nesting and the inflow conditions. For further details on the implementation of the turbulence recycling method see Maronga et al. (2015).

Further, in order to avoid persistent streaks in the $u$-component, which may develop in neutrally-stratified flows and will
be recurrently recycled in case of vanishing $v$-component, we shifted the recycled turbulent signals along the $y$-direction at the inflow boundary, following Munters et al. (2016). At the eastern outflow boundary we set a radiation boundary condition (Miller and Thorpe, 1981). The root domain was initialized with three-dimensional data recursively copied from the precursor run, while the child domain was initialized with data obtained from the parent. We used an isotropic grid spacing of 4 m and 2 m within the root and the nested child domain, respectively. The cuboid and the smooth-hill case also encompass a third
domain with 1 m resolution (two-stage nesting).

In order to evaluate the effect of the nesting, we performed additional non-nested reference simulations with 4 m and 2 m grid spacing. However, due to its high computational demands, a 1 m non-nested reference simulation was not performed. The simulated time of the neutrally stratified simulations ranged from 4 to 7 hours. Data analysis started after 2 hours of simulated time. When spectral analysis was performed (homogeneous flat case), the time step was held constant at 1.0 s for
that simulation.

### 4.2.1 Neutrally stratified boundary-layer flow over flat terrain

Figure 9 shows an instantaneous horizontal cross-section of the $u$-component for the nested simulation. As typical for a neutrally stratified boundary layer, elongated streak-like structures can be observed (Hutchins and Marusic, 2007; Hutchins et al., 2012). These elongated structures preserve their size and amplitude when entering the child domain from the left and exiting
to the right.

Figure 10 shows horizontal profiles of the time- and $y$-averaged friction velocity $u_*$ within the child domain and the corresponding coarse- and fine-grid reference cases. In the coarse and fine reference cases $u_*$ is constant along the $x$-axis indicating that the flow is in equilibrium with the surface friction. In the coarse-grid simulations $u_*$ shows slightly higher values compared to the fine-grid reference simulation, even though the prescribed surface roughness is identical in all simulations. This suggests
that the flow in the coarse-grid simulations sees a slightly rougher surface, which we attribute to the less accurate representation of the vertical near-surface gradients of the wind profile compared to the fine-grid simulation. When the flow enters the child domain, the coarse-grid inflow wind profile is not in equilibrium with the surface friction any more and the near-surface flow decelerates, indicated by the higher values of $u_*$ near the child inflow boundaries. With increasing distance to the inflow boundary, $u_*$ rapidly decreases and reaches a minimum with lower values compared to the reference cases, until it increases
again reaching a secondary maximum and then asymptotically approaches a constant value, which is similar to the value of the fine-grid reference case in the grid-spacing ratio cases 2 and 3. However, at least for the given model domain size, $u_*$ does not approach the fine-grid solution in grid-spacing ratio case 4 but still exhibits higher values. This kind of spatial oscillation of $u_*$, which indicates an alternating deceleration and acceleration of the near-surface flow along the $x$-direction, shows that the

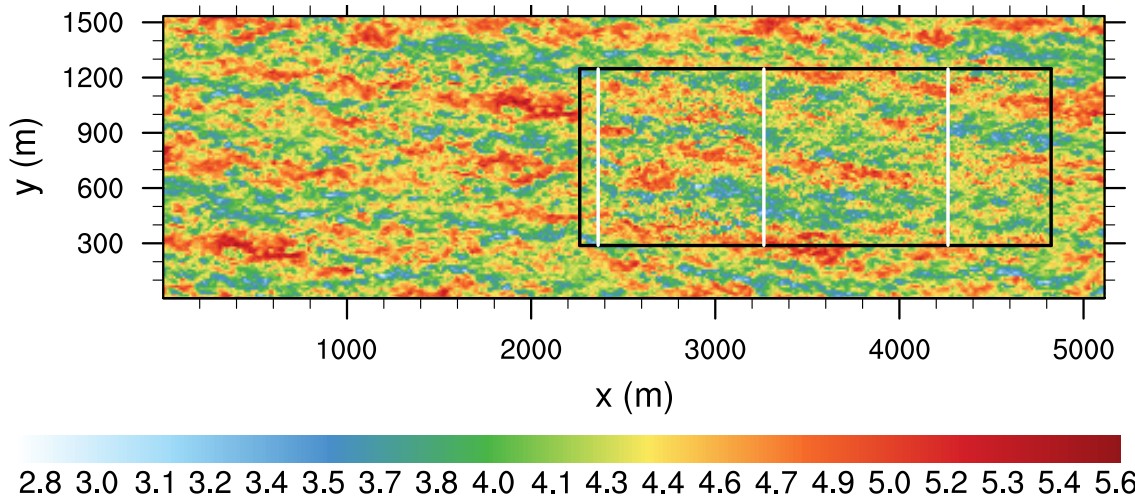

**Figure 9.** Instantaneous horizontal cross-section of the $u$-component in $\mathrm{m\,s}^{-1}$ at $z = 40$ m. The black box indicates the location of the child domain. The white solid lines indicate the $x$-locations where the profiles shown in Fig. 11 are averaged over the $y$-direction.

surface-momentum exchange in the child domain needs a sufficiently large development length. For grid-spacing ratio case 2
the required fetch length is of at least 1 km to adjust to the fine-grid resolution. With increasing grid-spacing ratio the amplitude of the spatial oscillation increases and the fetch length becomes longer, and, as in grid-spacing ratio case 4 even exceeds the model domain size.

Figure 10 shows that $u_*$ gradually adjusts to the fine-reference value, at least for grid-spacing ratio cases 2 and 3. This is in contrast to Moeng et al. (2007), who revealed a friction velocity bias between parent and child in their neutrally-stratified
simulation when employing grid-size dependent SGS model, which is the case also in this study.

Figure 11 shows time- and $y$-averaged profiles of the horizontal wind speed within the child domain for different grid ratios, taken at different distances downstream of the inflow boundary, indicated by the white solid lines in Fig. 9. At a distance of 100 m the profiles in the child domains agree well with the fine-reference profile within the lowest 10 m. Even though the surface-momentum exchange is still not in equilibrium at that position (see Fig. 10), one could already conclude from the
near-surface wind profiles that the flow has already been adapted to the finer grid resolution. However, further above, the wind profiles of the child model still deviate from the fine-reference solution and are closer to the coarse reference profiles. This is especially obvious for the grid-spacing ratio case 4, where the wind profile shows a discontinuity at a height of $z = 20$ m. With increasing distance from the child inflow boundary, the child-profiles gradually adjust to the fine-reference simulation, while at a distance of 2000 m the child profiles agree with the fine-reference solution, except for the grid-spacing ratio case 4 which
still deviates from the fine-reference solution.

In order to further analyze the flow adjustment within the child domain, we computed resolved-scale turbulent kinetic energy (TKE) spectra at different distances from the child inflow boundary. The spectra were calculated from time-series of the three velocity components that were sampled at different locations within the domain. The final spectra were then obtain by

averaging individual spectra over all locations with identical distance to the inflow boundary, assuming that the flow is parallel
to the $x$-axis. Figure 12 shows TKE spectra obtained from the child domain and for the corresponding reference simulations.
At low frequencies (large wave numbers), the spectra look quite similar and no obvious differences to the fine- and coarse-
reference spectra can be observed, indicating the grid nesting does not induce any larger-scale oscillations which propagate
through the model domain. At higher frequencies, however, especially the near-inflow boundary, the child spectra differ from
the fine-reference spectra and resemble more the corresponding coarse reference spectra. With increasing distance from the
inflow boundary, the spectral properties gradually adjust to those of the fine-reference case, while at a fetch of 500–1000 m
almost no differences can be observed any more at that height level.

In contrast to a buoyancy-driven boundary layer, the flow in a purely shear-driven boundary layer requires a sufficiently
large development distance to adjust to the finer grid resolution in terms of spectrally similar conclusion. However, a purely
shear-driven flow over a flat homogeneous surface can certainly be considered as an extreme case in terms of flow adjustment,
as the vertical turbulent exchange, which is primarily driven by surface-roughness induced shear, is rather low compared to less
idealized flows over non-flat terrain or with obstacles included. Hence, we expect that the required fetch length may decrease
for rougher surfaces and more complex surface geometries.

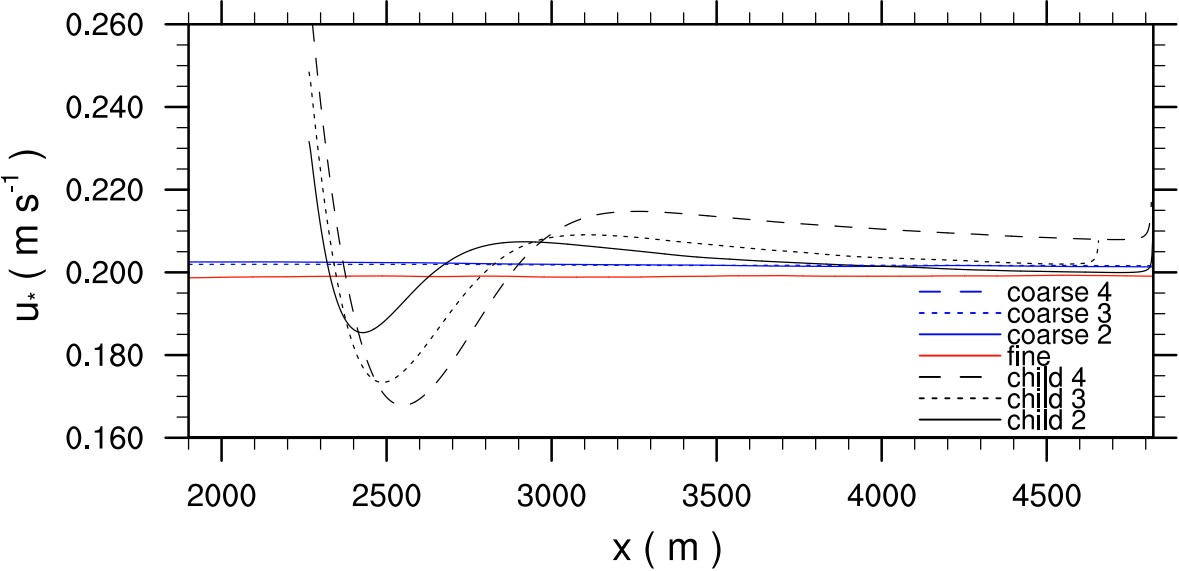

**Figure 10.** Two-hour time- and $y$-averaged horizontal profiles of the friction velocity for the grid-spacing ratio of 2, 3, and 4, as well as the
corresponding coarse- and fine-grid reference simulations.

### 4.2.2 Neutrally stratified boundary-layer flow over a smooth three-dimensional hill

The hill case is studied to compare flow statistics against the wind-tunnel observations conducted by Ishihara et al. (1999),
who sampled data at different up- and downstream locations along the centre hill axis. This flow is simulated using a two-stage

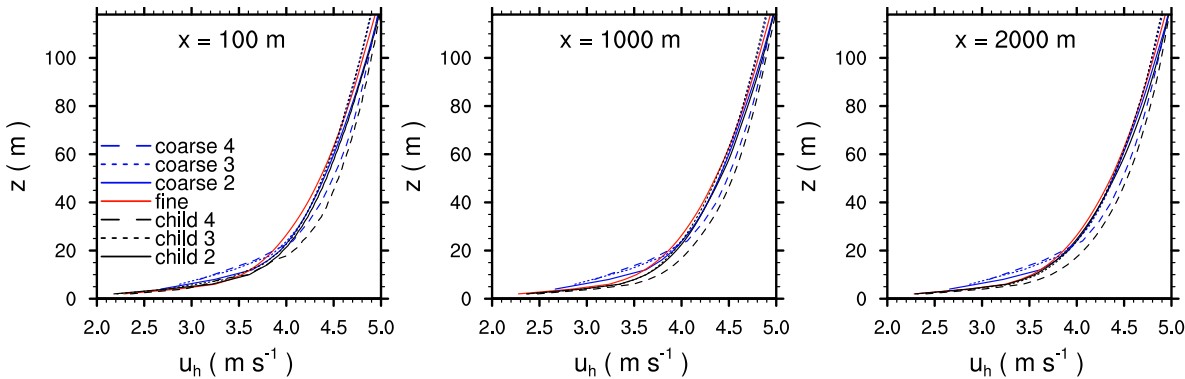

**Figure 11.** Two-hour time- and $y$-averaged profiles of the horizontal wind speed for the grid-spacing ratio of 2, 3, and 4, taken at different distances downstream of the inflow boundary, indicated by the white solid lines in Fig. 9. Also, corresponding time- and $y$-averaged profiles from the fine and coarse reference simulations taken at the same locations are shown.

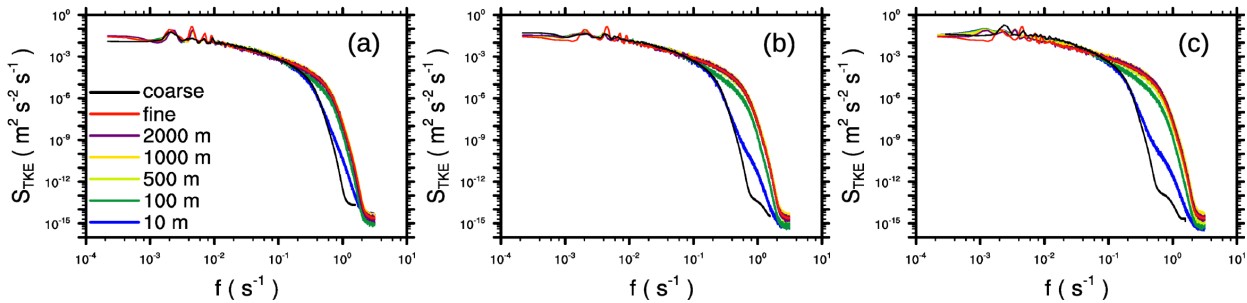

**Figure 12.** Frequency spectra of the resolved-scale TKE taken at different sampling locations downstream of the inflow boundary for the neutrally-stratified boundary layer at $z = 40$ m, for grid-spacing ratio a) 2, b) 3, and c) 4. TKE-spectra for the fine- and the corresponding coarse-grid reference simulations are also shown.

nesting configuration in which a second child domain with 1 m resolution is placed within the first child domain with 2 m resolution.

The terrain height of the smooth three-dimensional hill is given by

$$z(x,y) = H \cos^2 \left( \frac{\pi \sqrt{(x - x_0)^2 + (y - y_0)^2}}{2l} \right), \tag{17}$$

with the hill height $H = 40$ m and a hill radius $l = 100$ m, while $x$ and $y$ indicate the location on the discrete grid and $x_0 = 2024$ m and $y_0 = 384$ m the location of the hill top with respect to the parent domain dimensions. Note that we up-scaled the hill dimension by a factor of 1000 with respect to the wind-tunnel model. Again, the parent domain size is $L_x \times L_y \times L_z = 5.1 \times 1.5 \times 0.32$ km$^3$ in the $x$-, $y$-, and $z$-directions, respectively. The child domain sizes are $L_x \times L_y \times L_z = 0.768 \times 0.384 \times 0.16$ km$^3$ and $L_x \times L_y \times L_z = 0.576 \times 0.288 \times 0.12$ km$^3$ for the first and second child domains, respectively. The upstream boundaries

of the first and second child domains are placed about 9.7 $H$ and 6.3 $H$ upstream of the hill top, respectively. The smooth hill geometry is approximated with the grid-following stair-step geometry in PALM due to its orthogonal grid arrangement and its topography description system, see (Maronga et al., 2015).

Figure 13 shows the mean flow field along the centerline of the three-dimensional hill for the nested child simulations as well as the fine- and coarse-grid reference simulations. Upwind of the hill the mean flow in the nested simulations agree with
630 the one in the fine and coarse reference simulation. In the coarse-grid reference simulation the re-circulation extends further downstream up to about 4.1 $H$ on the lee side of the hill compared to the 2 m nested and fine reference simulation which both show a re-circulation that extends up to about 3.75 $H$ downstream of the hill top. With further increasing the grid resolution to 1 m the re-circulation zone further shortens to about 3 $H$. Figures 14 and 15 show the corresponding standard deviations of the $u$- and $w$-components sampled at different locations along the centerline of the hill. Upwind of the hill the standard deviations
of the $u$- and $w$-components agree with the observations and show no significant difference between the simulations. However, leeward of the hill at 1.25 $H$, the LES underestimates the standard deviations of the $u$- and $w$-components in the 2 m and 4 m simulations, which is most pronounced in the coarse-grid reference simulation, while the profiles in the 1 m child domain of the nested simulation agree fairly well with the measurements. Further downstream, the coarse-grid reference run still slightly underestimates the observed standard deviations, while the 2 m child domain result and fine-grid reference result slightly
overestimate the standard deviations, which is in agreement with results from the EPFL-LES model presented in Diebold et al. (2013) who employed a similar fine grid resolution for this hill flow. With 1 m grid spacing the standard deviations further downstream are still slightly overestimated, though their vertical shape and amplitude is captured quite well. In order to provide a more quantitative measure, Table 1 provides the root-mean square deviation (RMSD) of the profiles shown in Fig. 14 and 15. RMSD is defined as

$$\mathrm{RMSD}\left(\psi\right) = \sqrt{\left\langle \left(\overline{\psi} - \overline{\psi}_{\mathrm{Ref}}\right)^2 \right\rangle}. \tag{18}$$

where $\psi$ is any prognostic variable from the considered solution and $\psi_{\mathrm{Ref}}$ refers to the corresponding measurement value. The RMSD-values indicate that the profiles converge towards the observations with increasing grid resolutions. Furthermore, the standard deviations from the 2 m child domain and from the fine-grid reference simulation show only marginal differences among each other, indicating that the nesting method and the fine-grid reference simulations lead to almost identical results.
Considering that the hilltop is placed only about 390 m and 250 m downstream of the child domain inflow boundary, this indicates that in more complex setups where topography is present, the adjustment fetch can become significantly smaller compared to purely flat terrain as discussed in Sect. 4.2.1.

### 4.2.3 Neutrally stratified boundary layer over a regular array of cubes

The final test case features a neutral atmospheric boundary layer flow over flat terrain which becomes incident with a staggered
pattern of cubical obstacles. The resulting flow scheme resembles urban canopy turbulence where the interaction between roughness elements and ABL turbulence is primarily resolved. Here, the cubical obstacle height is $H = 40$ m. The distance between the obstacles is $3H$ in the $x$-directions and $1H$ in the $y$-direction.

**Table 1.** Root-mean square deviation (RMSD) of the simulated $u$- and $w$-standard deviations evaluated against the corresponding wind-tunnel measurements at the positions shown in Fig. 14 and 15. The RMSD is evaluated at all respective heights of the observations and is averaged over all heights and profile locations.

| Simulation | RMSD ($\sigma_u$) | RMSD ($\sigma_w$) |
|---|---|---|
| REF 4 m | 0.043 | 0.078 |
| REF 2 m | 0.022 | 0.054 |
| Child 2 m | 0.026 | 0.055 |
| Child 1 m | 0.015 | 0.039 |

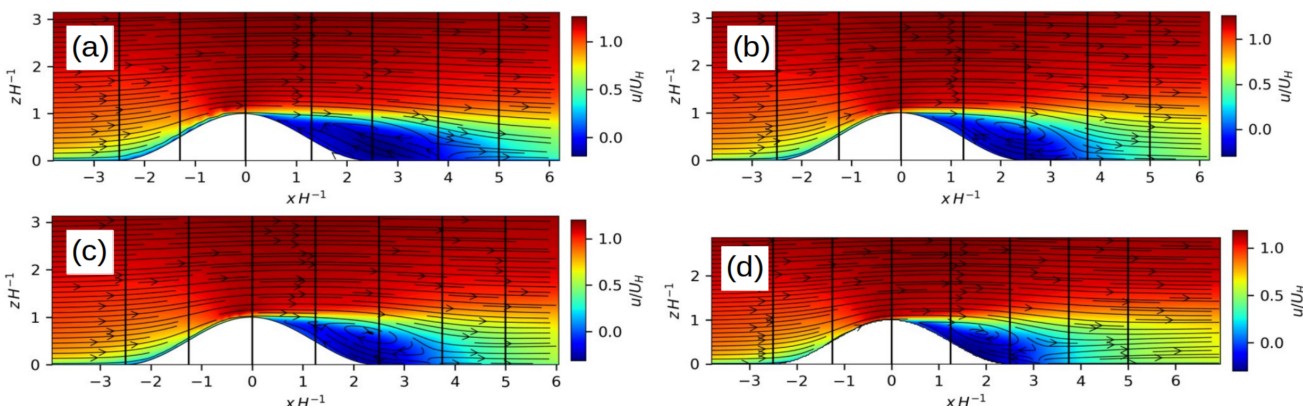

**Figure 13.** Two-hour time-averaged vertical cross-section of the $u$-component field (colored contours) as well as the $u - w$ mean flow field (vector arrows) displayed along the centerline of the three-dimensional hill for a) the $4$ m reference simulation, b) the $2$ m reference simulation, c) the $2$ m nested child simulation, and d) the $1$ m nested child simulation. Vector arrows as well as the $u$-component are normalized with the reference wind speed taken at $z = H$ upwind of the hill. The ordinate and the abscissa are scaled with the hill height $H$. Note, the abscissa is centered at the hill top. The black vertical lines indicate the positions of the profiles displayed in Fig. 14 and 15. Note that the ordinate in d) is constrained by the vertical dimension of the $1$ m child domain. Moreover, note that the $x$-dimension in c) and d) do not show the entire child domain extents but cut out minor parts at the left and right.

To demonstrate the flexibility of the nesting implementation, we carried out simulations with two different nested configurations illustrated in Fig. 16. The first (v1) case features a single child domain while the second (v2) case contains a two-stage nesting system where a second child domain is nested within the first. In the latter configuration, the first child acts as a parent for the second child domain. The isotropic grid spacing is $4$ m in the root domain, $2$ m in the second level nest (first child) and $1$ m in the third level nest (second child). (Note, that the implementation does allow child locations to be selected such that their domain boundaries intersect with the obstacles.) The two example configurations represent nesting applications designed to meet different levels of accuracy demands. The v2 configuration is set to resolve the transition effect at the leading edge

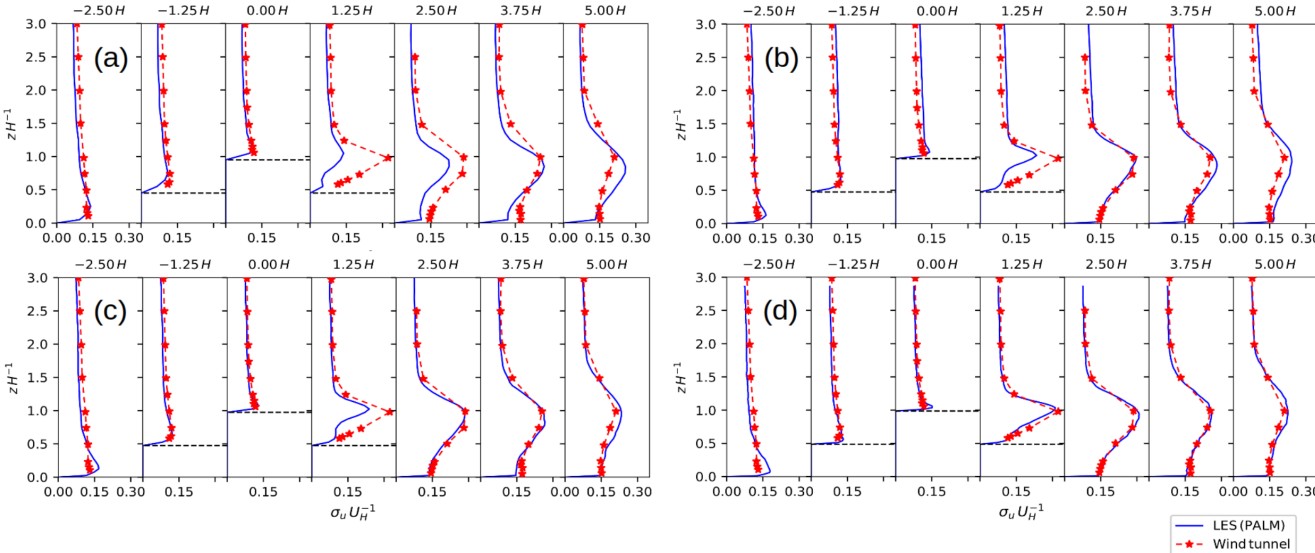

**Figure 14.** Two-hour time-averaged vertical profiles of the standard deviation of the $u$-component for the LES and the observed wind tunnel flow, for a) the $4$ m reference simulation, b) the $2$ m reference simulation, c) the $2$ m nested child simulation, and d) the $1$ m nested child simulation. The ordinate is scaled with the hill height $H$. The standard deviation is normalized with the reference wind speed taken at $z = H$ upwind of the hill. The black dashed horizontal lines indicate the discrete height of the surface at the sampling location. Note that the vertical dimension of the $1$ m nested child domain does not cover the entire vertical range of the measurements so that some data points from the LES are missing on the normalized height coordinate in d).

of the cube canopy and to capture the blunt-body wake interactions in sufficient detail within the center region of the cuboid canopy.

First, in the context of obstacle-resolved LES, we motivate the employment of an optional *canopy-restricted* (CR) anterpolation strategy introduced in Sect. 3.5. For this purpose, consider Fig. 17 showing an instantaneous horizontal cross-section of vorticity vector magnitude at $z = 0.9H$ height for configuration v2. The image is focusing on a region where all domains with different resolutions are visible.

The visualization indicates the strength and spatial structure of the resolved turbulent eddies and how they are affected by grid resolution. The differences are significant. In such obstacle-resolving LES, the increased grid resolution has the ability to alter the flow solution to such a degree that the anterpolation introduces details to the coarser parent which are inconsistent with the rest of the parent's flow solution. Particularly with blunt-body obstacle canopy flows, this discrepancy is clearly manifested as a locally changing resultant pressure drag (caused by the obstacles) within the anterpolated domain. To inspect this, we compute the resultant pressure drag coefficient

$$C_{Fp} = \frac{2}{\rho U_{ref}^2 S_{ref}} \left( F_{p,x}^2 + F_{p,y}^2 \right)^{1/2} \tag{19}$$

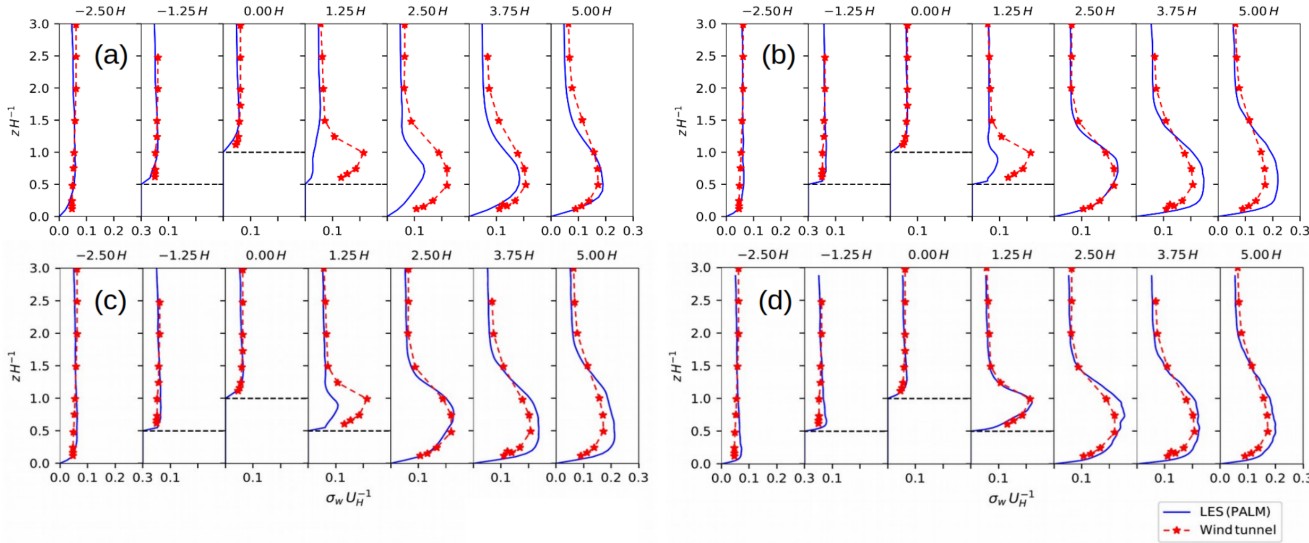

**Figure 15.** Two-hour time-averaged vertical profiles of the standard deviation of the $w$-component for the LES and the observed wind tunnel flow, for a) the 4 m reference simulation, b) the 2 m reference simulation, c) the 2 m nested child simulation, and d) the 1 m nested child simulation. The ordinate is scaled with the hill height $H$. The standard deviation is normalized with the reference wind speed taken at $z = H$ upwind of the hill. The black dashed horizontal lines indicate the discrete height of the surface at the sampling location. Note that the vertical dimension of the 1 m nested child domain does not cover the entire vertical range of the measurements so that some data points from the LES are missing on the normalized height coordinate in d).

for the differently coupled simulations. In Eq. 19 $u_{ref} = \langle \overline{u} \rangle |_{z=1.25H}$ is the reference wind speed, $F_p$ is the resultant pressure force exerted on the cubes obtained by integrating the pressure over vertical walls, $\rho$ is the density of air and $S_{ref}$ is the accumulated frontal area of the cubes. The results are listed in Table 2, which makes evident the drastic difference between the values for the coarse reference and the two-way coupled parent ($C_{Fp}$[Coarse] vs. $C_{Fp}$[Root]: two-way). This large difference arises as the anterpolated solution within the obstacle canopy introduces a large-scale disturbance to the parent solution giving rise to unphysical secondary effects. These effects, in turn, lead to complicated feedback systems in the two-way coupled solutions whose realizations become dependent on the chosen nesting configuration.

This problematic behavior is significantly abated by adopting the CR anterpolation strategy setting here the vertical threshold at $1.25H$ via experimenting. This CR anterpolation allows the parent and child flow fields to become strongly coupled while minimizing global inconsistencies in the parent solution. While all the child domain solutions over-predict the pressure drag, the two-way CR solution yield $C_{Fp}$[Child 2] values that are closest to the fine reference.

To further evaluate the nesting performance, we exploit root (normalized) mean square difference (RNMSD or RMSD) and fractional bias (FB) as comparison metrics (see, Britter and Hanna, 2003) evaluated over successive $xy$-planes to assess the effectiveness of the nesting approach in obstacle-resolving LES cases. RNMSD and RMSD provide a measure of mean difference that is composed of random scatter and systematic bias whereas the fractional bias (FB) yields a specific measure

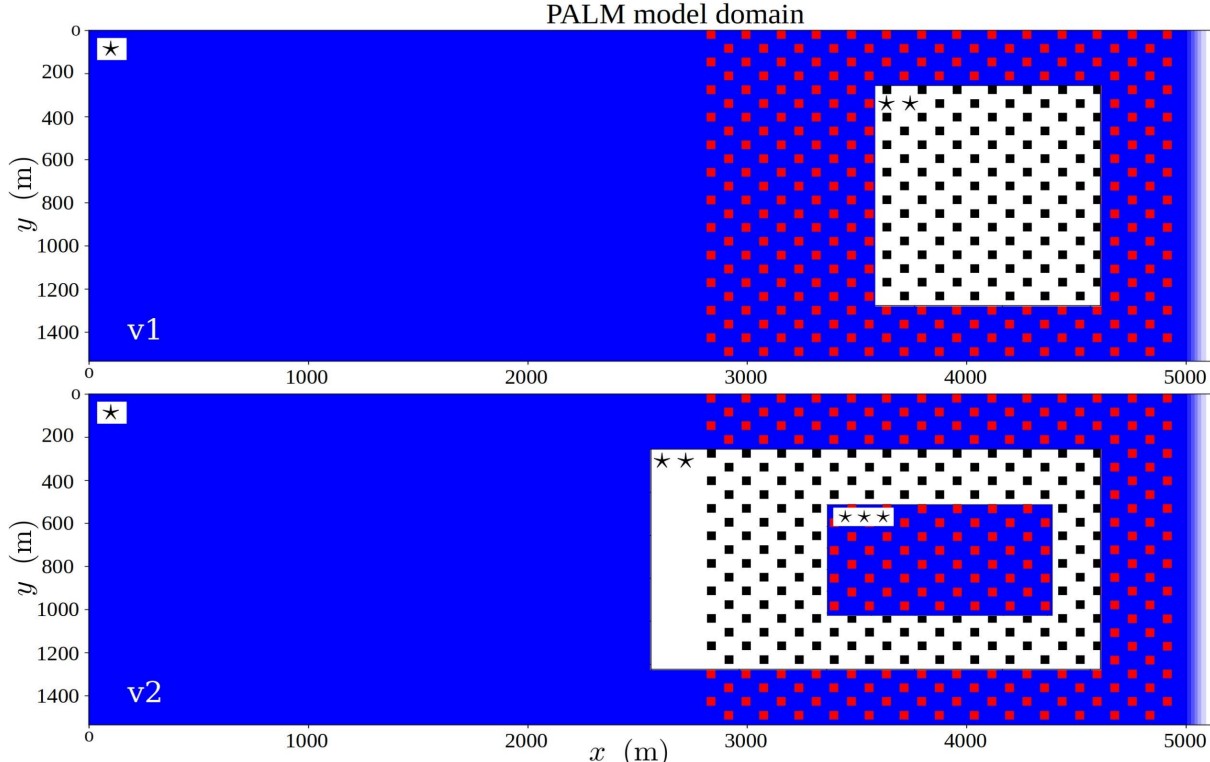

**Figure 16.** An overview of the cubical-obstacle case layout. The obstacles are cubes with 40 m sides. The figure displays two nested arrangements: version 1 (v1) featuring a root domain and a secondary nest domain whereas version 2 (v2) also includes a tertiary nest domain embedded within a larger secondary nest. The root and nested domains are indicated with (★), (★★), and (★★★) respectively in the upper left-hand-corner of each domain. The first child domain is displayed with white background for better visualization.

**Table 2.** Resultant pressure force coefficients $C_{Fp}$ evaluated over Child 1 (★★) domain shown in Fig. 16 (v1). Results for parent and child solutions are reported for one-way, two-way and two-way canopy-restricted (CR) methods, where the latter is a modified two-way coupling approach where the anterpolation is restricted (i.e. not allowed) within the obstacle canopy.

| Version 1 | | | | Version 2 | | |
|---|---|---|---|---|---|---|
| | one-way | two-way | two-way CR | | one-way | two-way | two-way CR |
| $C_{Fp}$[Root] | 0.592 | 0.735 | 0.549 | | 0.790 | 1.017 | 0.785 |
| $C_{Fp}$[Child 1] | 0.602 | 0.599 | 0.594 | | 0.828 | 0.859 | 0.826 |
| $C_{Fp}$[Fine] | | 0.583 | | | | 0.808 | |
| $C_{Fp}$[Coarse] | | 0.592 | | | | 0.790 | |

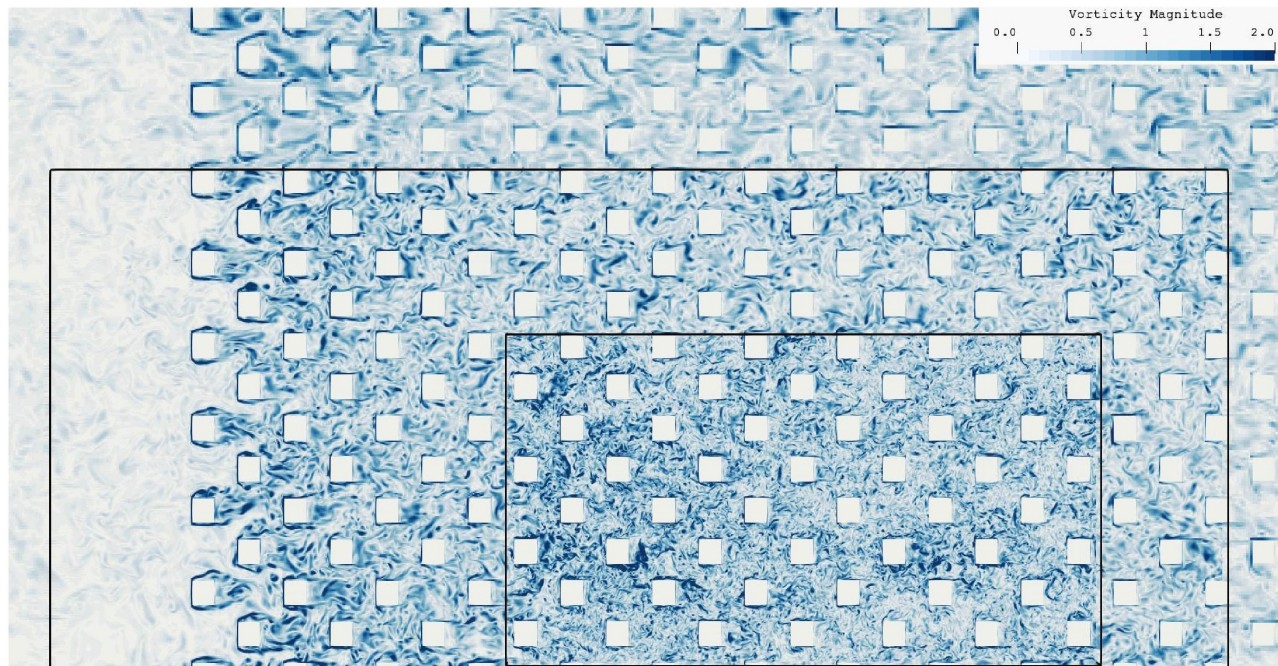

**Figure 17.** Instantaneous close-up view of vorticity magnitude ($s^{-1}$) on $xy$-plane at elevation $z = 0.9H$ for the v2 case with two nested domains. Black lines indicate the bounds of the first and the second nest. Note, only parts of the domain extents are displayed.

for the systematic bias between the two solutions. The RMSD metric is defined by Eq. (18) while RNMSD and FB are defined as

$$\text{RNMSD}(\psi) = \sqrt{\frac{\left\langle \left( \overline{\psi} - \overline{\psi}_{\text{Ref}} \right)^2 \right\rangle}{\left\langle \overline{\psi} \right\rangle \left\langle \overline{\psi}_{\text{Ref}} \right\rangle}} \tag{20}$$

$$\text{FB}(\psi) = \frac{\left\langle \overline{\psi} \right\rangle - \left\langle \overline{\psi}_{\text{Ref}} \right\rangle}{0.5 \left( \left\langle \overline{\psi} \right\rangle + \left\langle \overline{\psi}_{\text{Ref}} \right\rangle \right)}. \tag{21}$$

where $\psi$ is a generic prognostic variable from the considered case, while here $\psi_{\text{Ref}}$ refers to the value from the reference simulation with 2 m resolution. RMSD is used instead of RNMSD in cases where the product of double-averaged quantities used for normalization approaches zero. Similarly, FB is only evaluated for the streamwise velocity component because other components yield a near-zero denominator which contaminates the metric. The evaluations are performed for 15 $xy$-planes within the child domain (zone ($\star\star$) in Fig. 16) which are equally spaced over the range $0 \leq z/H \leq 1.5$. We have excluded 128 m and 64 m wide development zones at the boundaries in the $x$ and $y$-directions respectively. When the coarse (4 m resolution) reference solution is compared to the fine (2 m resolution) reference, the coarse solution is interpolated onto the fine grid before the comparison metrics are evaluated.

Both model variants v1 and v2 are included in the analysis to demonstrate how the size and placement of the child domains effect the metrics and also to illustrate the possibility to employ a cascade of nested domains. Although no comparison metrics are presented for the second child solution featuring 1 m resolution, its influence is embedded in the solution of the first child.

The RNMSD and RMSD profiles for the velocity components and their variances depicted in Figs. 18 and 19 lay bare the effectiveness of the presented nesting system and reveal the added benefit of the CR anterpolation. While all the coupling ap-
710 proaches succeed in significantly reducing the discrepancy compared to the fine reference, the conventional two-way coupling exhibits the most pronounced level of deviation. The FB results in Fig. 20 indicate that the two-way coupled solution also contains the most systematic deviation, which is conform with the pressure drag results.

The one-way coupling approach performs consistently better than the unmodified two-way coupled in all metrics, but it is also associated with a systematic bias that is larger than the value by coarse reference. However, if the modest systematic shift
in streamwise velocity can be accepted, the one-way coupling offers a cost-effective nesting coupling approach (see Sect. 4.3 for performance measures). Nonetheless, the results conclude that the introduced CR anterpolation approach presents the most recommended coupling strategy for obstacle-resolving LES as it provided the best metrics in every category.

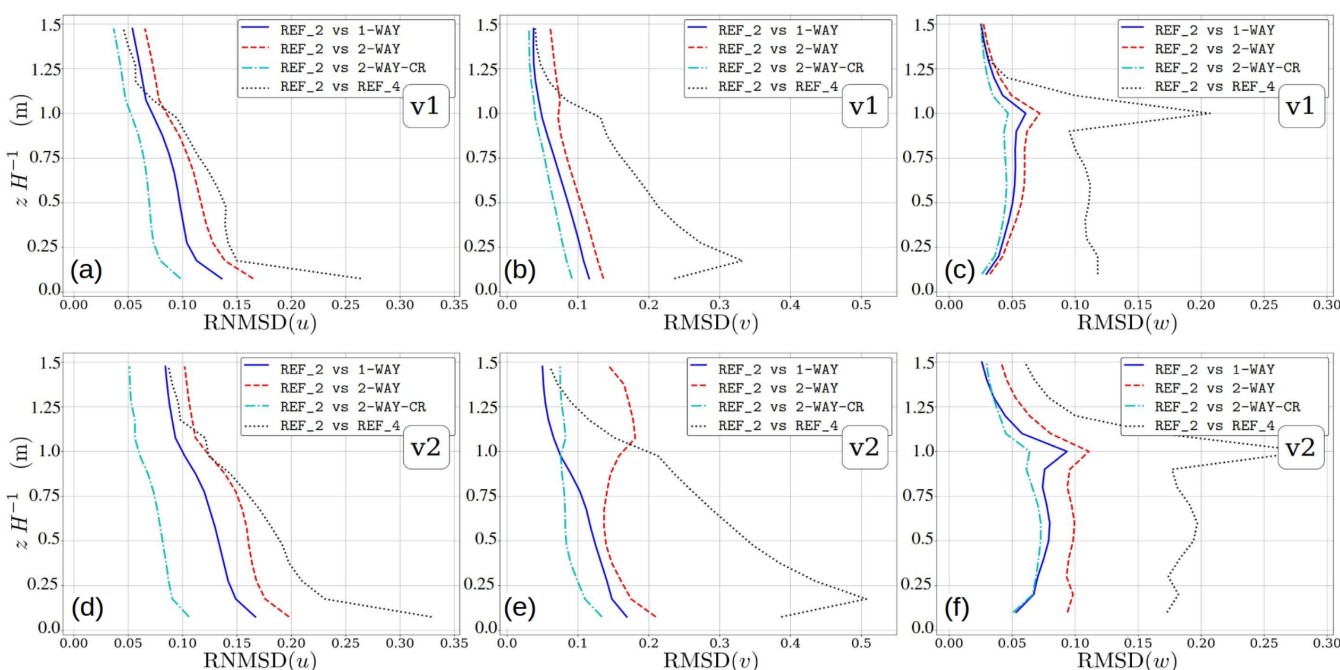

**Figure 18.** Vertical distributions of root (normalized) mean square difference (RNMSD or RMSD) of velocity components for configuration v1 a-c) and configuration v2 d-f). The metrics are evaluated for 15 horizontal planes between the range $0 \leq z/H \leq 1.5$.

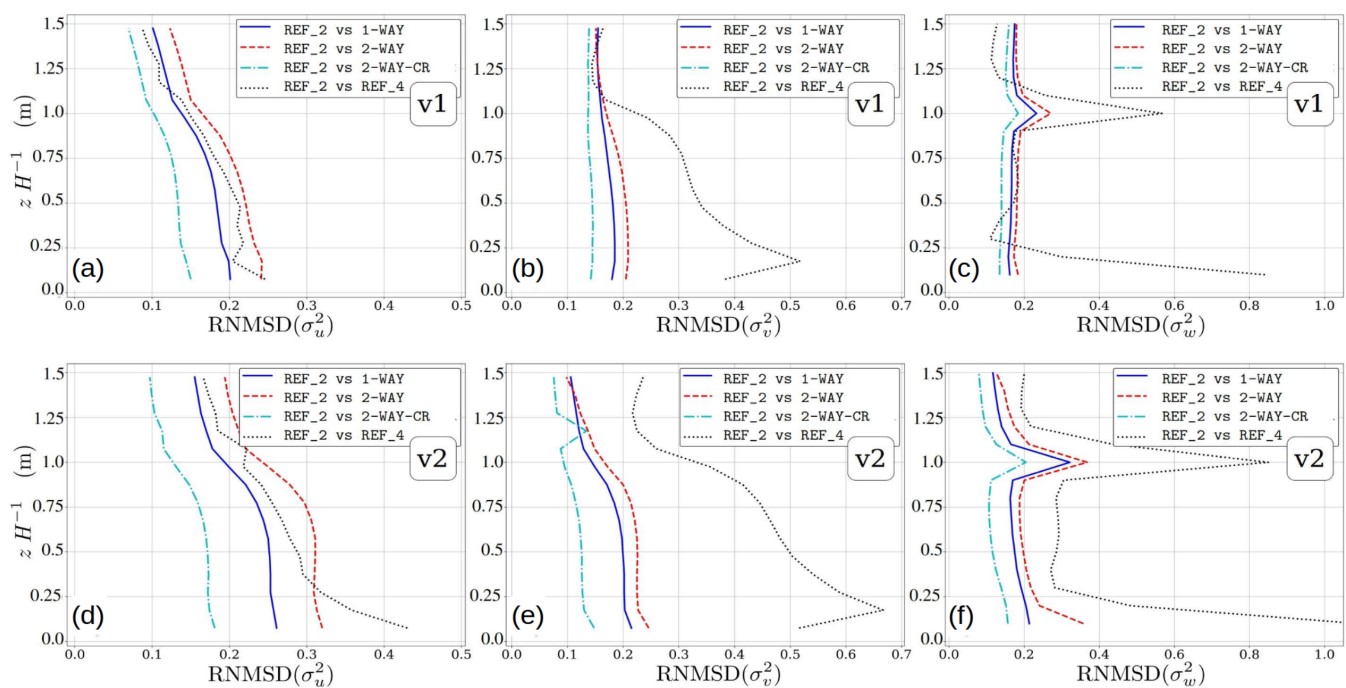

**Figure 19.** Vertical distribution of RNMSD of the horizontal velocity variances for configuration v1 a-c) and configuration v2 d-f). The metrics are evaluated for 15 horizontal planes between the range $0 \leq z/H \leq 1.5$.

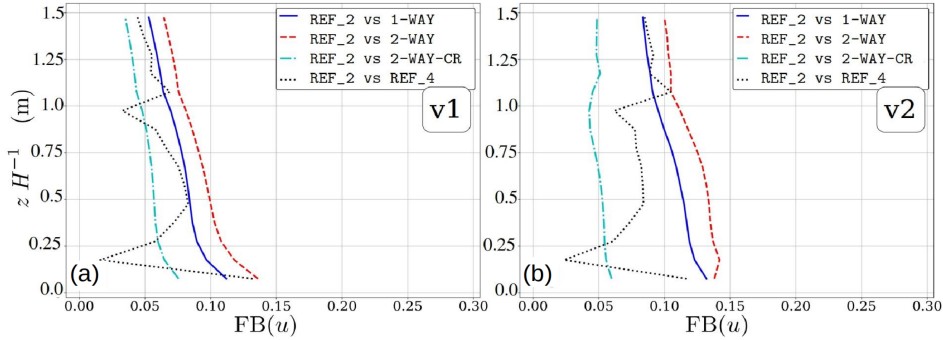

**Figure 20.** Fractional bias (FB) values evaluated over 15 $xy$-planes between the range $0 \leq z/H \leq 1.5$, for a) v1 and b) v2.

## 4.3 Performance issues

Table 3 gives an overview of the consumed CPU time $T_{\mathrm{CPU}}$, wall-clock time $T_{\mathrm{wall}}$ and some performance measures in the nested as well as fine- and coarse-grid reference simulations for the hill and the cube case simulations. $T_{\mathrm{CPU}}$ is here defined simply as $N_{\mathrm{procs}}T_{\mathrm{wall}}$ and $N_{\mathrm{procs}}$ is the number of parallel processes. As a rule of thumb, doubling the resolution leads to an increase in CPU time by approximately a factor of 16 (when the numerical time step is determined according to the Courant-

Friedrichs-Lewy criterion (CFL criterion). This can be observed comparing the coarse- and fine-grid reference simulations. Compared to the fine-grid reference simulations, the nested simulations consumed significantly less CPU time (up to 80% reduction) while increasing the computational cost by factors of 3.8 and 3.4 in hill and cube canopy cases compared to the coarse-grid reference. The direct overhead due to the nesting, i.e. the fraction of the CPU-time used by all the nesting-related operations during the time-stepping is reasonably small in these tests, only from 2% to 16%. Also the CPU time per grid point per time step increases only quite moderately due to the nesting. These figures differ surprisingly much between the hill and the cube-array cases. This is most likely due to the fact that these cases were run on different computer systems. Although these factors depend on the child domain size, these tests make evident that the nesting technique can significantly reduce the computational cost, while yielding results that closely adhere to the non-nested fine-resolution simulation.

Due to the inter/anterpolation and the accompanied inter-model data transfer, the nesting itself consumes CPU time. In our tests the workload with respect to the number of grid points treated by a processor element was equal among the parent and the child simulation. With this optimal configuration, the two-way nesting consumed about 10-16% of the CPU time in our tests, while it consumes only about 2% in the one-way nesting. This suggests that most of the CPU time taken by two-way nesting is consumed in the anterpolation and the associated child to parent data transfer.

It is important to bear in mind that if the workload between child and parent processes is not well balanced, the faster processes need to wait before the data-transfer can start until the slower processes reach that point, reducing the computational efficiency of the nesting.

**Table 3.** Total number of grid points $N_{\mathrm{gp}}$, number of parallel processes $N_{\mathrm{procs}}$, required CPU and wall-clock times and other performance measures for the hill and the cube array v1 test cases. Here $N_{\Delta t}$ is the number of time steps taken during the run.

| Case | $N_{\mathrm{gp}}$ | $N_{\mathrm{procs}}$ | $T_{\mathrm{CPU}}$, h | Nesting overhead | $T_{\mathrm{wall}}$, h | $T_{\mathrm{CPU}}/(N_{\Delta t}N_{\mathrm{gp}})$, µs |
|---|---|---|---|---|---|---|
| Hill coarse | 39 321 600 | 512 | 220 | – | 0.4297 | 0.658 |
| Hill fine | 314 572 800 | 2048 | 3910 | – | 1.909 | 0.692 |
| Hill nest | 45 219 840 | 768 | 831 | 16% | 1.082 | 1.085 |
| Cubes coarse | 39 321 600 | 120 | 855 | – | 7.130 | 1.290 |
| Cubes fine | 314 572 800 | 480 | 12 900 | – | 26.88 | 1.166 |
| Cubes v1 nest one-way | 56 098 816 | 184 | 2580 | 2.1% | 14.02 | 1.427 |
| Cubes v1 nest two-way | 56 098 816 | 184 | 2910 | 12% | 15.82 | 1.636 |

## 5 Conclusions and future outlook

This article documents and evaluates an online LES-LES nesting scheme implemented into the PALM model system 6.0. The nesting system relies on the post-insertion approach and features both one-way and two-way coupling approaches. We give a detailed description of the model's relevant technical, algorithmic and numerical aspects and provide evidence for the accuracy

gains the method introduces with a dramatically reduced computational cost compared to globally refined grid resolution. Particularly in urban boundary layer studies requiring obstacle-resolving LES, the nesting approach has proven essential.

The implementation of this three-dimensional nesting system is based on two-level parallelism involving inter-model and intra-model parallelization using MPI. This enables our nesting implementation to flexibly support multiple child domains which can be nested within their parent domain either in a parallel or recursively cascading configuration. All solutions involved within the nested simulation are advanced using a globally synchronized time step whereas the coupling between each parent/child pair is performed with interpolation (parent to child) and anterpolation (child to parent) operations.

The nesting method is evaluated by performing a series of numerical experiments with an objective to demonstrate that the refined child solution (nested within a coarser parent) approaches the non-nested reference solution obtained by employing fine resolution globally.

The first test case features horizontally homogeneous convective boundary layer (CBL) with no mean wind. In this case, first and second order boundary-layer statistics are well captured in the child domain and are closely comparable to non-nested high-resolution reference statistics. Further, due to the local nature of turbulence production and the weak advection from parent into the child, the flow statistics show almost no dependence on the distance to the child boundaries. However, in case of several hours long averaging times we found that a nonphysical secondary circulation develops although the surface heating is homogeneous. We hypothesize that this secondary circulation is an inherent consequence of the spatially changing description of flow physics in the parent and child solutions. Even though we demonstrated this issue with a rather idealized setup using unrealistically long averaging time and such nonphysical circulation are probably minimized in less idealized simulations, e.g. with wind, a diurnal cycle and realistic terrain surfaces, we believe that this should be kept in mind when applying the nesting system to CBL-problems.

The second test case simulated neutrally stratified boundary layer flow over flat terrain. The nested simulations reveal that the flow solution within the child domain must undergo a development phase, as the flow solution adjusts to the higher resolution, before reaching equilibrium state again. The required development length depends on the grid-spacing ratio between parent and child. However, a purely shear-driven flow over a homogeneous flat terrain can be considered as an extreme scenario with respect to the development length of turbulence, while in cases with more complex surface geometry the flow adapts within shorter development distances. Beyond the development distance, the child solution for grid-spacing ratios of two and three agree well with the non-nested fine-reference solution, but in case of grid-spacing ratio of four the results clearly deviate from the fine-reference solution.

The third numerical experiment featured boundary layer flow (similar to second test case) over a smooth three-dimensional hill. This test case also exploits wind tunnel measurements to strengthen the nesting model evaluation. In this case, the flow statistics in the windward and the leeward part of the hill are almost the same as in a fine-reference simulation and agree well with wind-tunnel observations presented in Ishihara et al. (1999).

The final test case examines a flow system where a fully developed boundary layer flow becomes incident with a staggered arrangement of cube-shaped obstacles. This flow scenario closely resembles an obstacle-resolving urban boundary layer flow situation. The case revealed that in two-way coupled simulations, the anterpolated child solution introduces discrepancies

within the parent domain which is manifested as elevated pressure drag within the anterpolated zone. This complication is remedied by introducing *canopy-restricted* anterpolation approach, where anterpolation is omitted within the obstacle canopy. By computing comparison metrics, root-normalized mean square difference and fractional bias, to quantify the difference between the fine reference and the nested solutions, the canopy-restricted two-way coupling is shown to be the best coupling strategy for obstacle-resolving LES studies.

**Future outlook**

Future development is planned to include the following tasks. Incorporation of PALM's Lagrangian particle model in the nesting system in order to enable Lagrangian dispersion studies in urban environments in such a way that particles can be transferred between parent and child domains depending on their position. Thus, the long-distance transport of e.g. pollutants, can be simulated in a coarse-resolution parent grid, while dispersion on the street-scale for specific locations can be simulated in a fine-resolution child domain. We note that this has been already implemented into PALM and is available to users, but further sensitivity tests with respect to the treatment of stochastic subgrid-scale particle speeds (Weil et al., 2004) are still pending. A thorough description and verification of the particle nesting will be published in a follow-up article.

Further, we note that the PALM model system 6.0 includes also a RANS (Reynolds-averaged Navier-Stokes) mode offering two different turbulence closures to calculate the eddy diffusivity, that are a TKE-$l$ and a TKE-$\epsilon$ closure according to Mellor and Yamada (1974, 1982). Besides the LES-LES nesting the nesting system is being extended to handle RANS-LES and RANS-RANS nesting, which require coupling of additional RANS-variables. Moreover, in a companion paper in this special issue we present a pure one-way off-line mesoscale nesting method in which the PALM model system 6.0 is nested into meso-scale models such as COSMO or WRF. This will allow modelling of meso-scale processes on a much larger coarse-grid domain as e.g. shown by Muñoz-Esparza et al. (2017), while concurrently focusing on fine-scale processes within certain areas using the present LES nesting approach.

Furthermore, to date, the timestep in all parent and child models is synchronized and restricted to the minimum of the time steps determined by each model independently using the CFL criterion. To our experience, the global timestep is often restricted by the flow around building edges where high wind speeds occur within the fine-grid child domains. Hence, we plan to implement a time-splitting into PALM where the parent and child models will be coupled only at the end of the parent timesteps. This would allow to run coarser-scale parent domains with larger time steps. Thus, computational time could be saved in the time-integration of the parent simulation as well as in the inter-model communication between parent and child.

**Appendix A: Technical realization**

**A1   General**

The nested model system is implemented using two levels of MPI communicators. The inter-model communication (communication between model domains) is handled by a global communicator using the one-sided communication pattern (remote

memory access, RMA). The intra-model communication (communication between subdomains within each model domain) is two-sided and it is handled using a 2-D communicator that has different color for each model. The intra-model communication system is the baseline parallelization of PALM (Maronga et al., 2015).

Data transferred from parent to child and from child to parent is always stored in the coarser parent-model grid in order to minimize the amount of data transfer. This means that the interpolations and anterpolations are always performed by the child.

For these purposes, children contain auxiliary arrays which follow the parent-grid spacings and indexing for each prognostic variable to be coupled covering the overlap domain plus necessary number of ghost-node layers.

## A2 Initialization

Mapping between each parent and child model domain decompositions as well as all the necessary index mappings are determined in the initialization phase and stored so that the coupling actions during the time-stepping are straightforward and

820 efficient.

Initial conditions for the root are set similarly to non-nested runs. The root then sends initial field data to its children which interpolate their own initial conditions from the data received from the root. Next the first-level children send their data to their children, if any, and so on. The basic interpolation subroutines for child boundary conditions operate only on the ghost nodes behind the child-model boundaries. Therefore a separate three-dimensional interpolation subroutine is implemented to

825 generate initial fields for all the nest domains from their parent-model fields. The same interpolation algorithm is used here as in the interpolations for child boundary conditions.

## A3 Modularization

The data transfer between parents and children is conducted by code contained by five specific fortran modules forming a module set called PALM Model Coupler (PMC). Calls to the PMC-subroutines are mostly made in PMC-interface module

(pmc_interface_mod.f90) such that only a small number of calls to the PMC-interface subroutines are needed within the baseline PALM code. This way, the changes to the baseline code were kept minimal. The PMC-interface module also contains subroutines for the nesting-related initialization actions, interpolation, anterpolation, child mass-balance forcing, etc.

## A4 MPI implementation

While reading the input namelists, the PALM root process checks if a namelist called "&nesting_parameters" is given in the

835 parameter input file PARIN. If not, subroutine called pmc_init_model resets all nesting-related parameters (coupling_layout etc.) and sets MPI_COMM_WORLD as the base global MPI-communicator comm_palm. The run then continues in standard way without nesting. If the namelist "&nesting_parameters" is found and correctly input, the root process of the root model distributes this information to all other processes via MPI_COMM_WORLD. Then, all the necessary nesting related parameters are determined and the base communicator is split into different colors for each model based on the model iden-

840 tification number. The term color means here that the communicator has the same name for all models (process groups),

but they are, however, individual communicators guaranteeing that communication of one model is not interfered by the others. The splitting is performed by calling MPI_COMM_SPLIT. Now each model has its own process group and associated individual base communicator color such that each model's internal communication is not visible to other models. After this, the mappings between models are determined. Each model, except the root model, identifies its parent model and creates an inter-communicator between the process groups of itself and its parent model. This is realized by calling MPI_INTERCOMM_CREATE. In the same way, each model identifies its all children if any, and creates inter communicators between the process groups of itself and all of its children. These inter communicators are only used to transfer setup data between the root processes of the parent and child models. For 3D model data transfers between parent and child, specific intra-communicator is created by merging inter-communicators of each processes within the remote process groups. This is made after pmc_init_model separately for child and parent models (note that a model may be both parent and child) in subroutines pmc_childinit and pmc_parentinit by calling MPI_INTERCOMM_MERGE. After the pmc-initialization, the run of each model goes as usual. Cartesian topology-based communicator comm_2d is created by each model from its color of the base communicator comm_palm using MPI_CART_CREATE.

The model internal communication is done in the usual way, e.g. by calling the boundary exchange routines. All data transfer between parent and child models is done within the PMC interface. For this communication MPI one sided communication (RMA) is used. An RMA window is opened on the parent side. To transfer data from parent to child, the parent fills the RMA window via local copy. After synchronization via MPI_WIN_FENCE, the child processes can fetch the data across the network with MPI_GET. While transfering data in the opposite direction, the child first transfers the data via MPI_PUT. After another MPI_WIN_FENCE call, the parent copies the data out of the RMA window into the local model data area.

**Appendix B: Thoughts on an alternative interpolation method**

Should higher interpolation accuracy across the boundaries be sought, the following considerations are relevant. As stated by Zhou et al. (2018), to satisfy the global flux-conservation requirement, one of the flux factors, either the advective velocity component or the advected variable, must be constant within the anterpolation cell. This implies zeroth-order interpolation. The other factor must be interpolated using any reversible interpolation scheme.

As stated above, the quadratic Clark and Farley (1984) scheme should not be used because it employs a stencil wider than the parent-grid cell which leads to problems with complex geometries. On the other hand, tri-linear interpolation has a favorable stencil width, but it is not suitable for the scheme as such is not reversible. However, linearly interpolated values $\widetilde{\phi}_{i,j,k}$ can be made reversible by introducing an additional correction $\phi_{i,j,k} = \widetilde{\phi}_{i,j,k} + \Delta\phi_{i,j,k}$ which guarantees the reversibility. The reversibility correction $\Delta\phi_{i,j,k}$ depends on the difference $\Delta\Phi_{I,J,K}$ between the original parent-grid value $\Phi_{I,J,K}$ and the value obtained by anterpolating the linearly interpolated values to the parent-grid node $I, J, K$ as

$$\Delta\Phi = \Phi - \widehat{\widetilde{\phi}}. \tag{B1}$$

$\Delta\Phi_{I,J,K}$ is a constant value within the parent-grid cell, hence a question arises: how to distribute the correction to the child-grid nodes $i, j, k$ such that $\widehat{\Delta\phi}_{I,J,K} = \Delta\Phi_{I,J,K}$? The simplest choice is $\Delta\phi_{i,j,k} = \Delta\Phi_{I,J,K}$, but this choice is not recommendable

in the cases of positive definite scalar variables as it could lead to negative values when $\Phi$ is close to zero. In principle this
problem could be avoided by weighting the local corrections in proportion to the local differences $\Phi_{I,J,K} - \widetilde{\phi}_{i,j,k}$ but this simply
reduces the method back to the zeroth-order baseline method. To make this approach useful, a more advanced technique to
distribute the correction ought to be developed. However, this is beyond the scope of the present work as stated in Sect. 3.4.4.

*Code and data availability.* The PALM model system is freely available at http://palm-model.org and distributed under the GNU General
Public Licence v3 (http://www.gnu.org/copyleft/gpl.html). However, the simulations presented in this document were performed using a
slightly modified code based on revision 4295. This modified source code (4295M) as well as the input files for the test runs are available at
https://doi.org/10.25835/0090593 (Hellsten et al., 2020). Numerous pre- and post-processing scripts are available at http://doi.org/10.5281/
zenodo.4005687 (Auvinen et al., 2020b).

*Author contributions.* Coordination of the study: AH, SR. Design and implementation of the inter-model communication: KK. Theoretical
considerations and implementation of the nesting interface: mainly AH with contributions from SR, MA, MS and BM. Simulations, post-
processing and analysis of model results: AH, MS, MA, BM, CK, FB, GT, NM. Drafting of the manuscript: AH, MA, MS, BM. Revision of
the manuscript: all authors.

*Competing interests.* The authors declare that they have no conflict of interest.

*Acknowledgements.* Test runs with PALM have been performed at the supercomputers of the North-German Super-computing Alliance
(HLRN), Germany, and CSC – IT Center for Science, Finland. This research was funded by Academy of Finland grant number 277664. BM
and MS were supported by the Federal German Ministry of Education and Research (BMBF) under grant 01LP1601 within the framework
of *Research for Sustainable Development* (FONA; https://www.fona.de). KK was supported by the Federal German Ministry of Economy
and Energy (BMU) under grant 0325719C. We wish to thank the anonymous referees as well as Mr. Sebastian Giersch at Leibniz Universität
Hannover and Dr. Jukka-Pekka Keskinen at Finnish Meteorological Institute for their help in improving the manuscript.

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
