# Peer review of "A nested multi-scale system implemented in the large-eddy simulation model PALM model system 6.0"

_Geoscientific Model Development, 2020_

## Short Comment (SC1) · 14 Sep 2020

In your text you write

"Our approach is to use Eq. (3) for the boundary normal velocity component u\_N and all scalar variables, [...]."

For the boundary-normal component it's clear but for the scalars you still do an average/interpolation ("phase 1 average", see the attached Figure). I thought equation (3) mean that the left parent grid cell value for scalars is directly taken to set the boundary values for the child grid boxes as it is done more or less for u. This is what equation (3)
suggest or am i wrong? Can it be that the phase 1 interpolation is not mentioned at all in the manuscript?

Additionally, i fear that further figures are necessary that show the interpolation scheme for one direction/child boundary (as in the attachement) and also the connection to the flux degradation (which points are actually used for calculating the fluxes) should be illustrated not only by words.

Kind regards, Sebastian Giersch

GMDD
Fig. 1.

---

## Referee Comment (RC1) · Anonymous Referee #1 · 13 Oct 2020

The authors developed an LES-LES nested multi-scale system in the PALM model. A key point is to keep unsteady turbulent behaviors in nesting LES-LES models. As far as I know, they developed the two-way coupling scheme in obstacle-resolving LES models for the first time in the world. They carefully evaluated them through the LES-LES numerical experiments on several types of turbulent flows such as NBL, CBL flows, building and hill flows. I strongly recommend publication of this manuscript after the following points are addressed.

Main comments 1. In page2 and line13: The authors mentioned "many numerical solution methods (e.g. finite-element and finite-volume methods)...". Recently, the

[Figure]

Lattice Boltzmann Method (LBM) has also come to be regarded as a useful tool and been applied to wind engineering field. Since the LBM has a merit of quite high-speed calculation, it can quickly conduct large-scale wind simulations even for urban areas resolved by a fine grid by massively parallel computing (e.g., Ahmad et al., 2017). Would you comment on the strength of the PALM model based on a finite difference method by comparison with the LBM method?

Ahmad et al.: Large-Eddy Simulation of the Gust Index in an Urban Area Using the Lattice Boltzmann Method, Boundary-Layer Meteorol, DOI 10.1007/s10546-017-0233-6, 2017.

2. In page3 and line6: The authors mentioned "according to our knowledge, WRF-LES is not applicable to blunt-obstacle resolving LES required for urban turbulence studies". the limitation of the WRF-LES nesting system. Wiersema et al. (2020) developed the WRF-IBM (immersed boundary method) to enable multiscale simulations over highly complex terrain with dynamically downscaled boundary conditions from the meso-scale to the building-scale. Although their approach is one-way nesting system, they successfully simulated turbulent flows in the urban central district by the WRF using the IBM method. The authors should refer the WRF-IBM study and comment on it.

WIERSEMA et al.: Mesoscale to Microscale Simulations over Complex Terrain with the Immersed Boundary Method in the Weather Research and Forecasting Model, Mon. Wea. Rev. (2020) 148 (2): 577–595.

3. In page13: It is difficult for readers to understand "Canopy-restricted anterpolation". Would you describe the idea of the aterpolation more clearly?

4. In page26: There seems to be different between the experiments and LES at a downstream position of 1.25H even for a fine reference simulation case. Is this due to the orthogonal grid system?

[Figure]

Minor comment 5. In page6: the left (1), right (1), south (2), north (2), → west (1), east (1),. . .? 6. In page26: Figure 13 and 14 → Figures 13 and 14

---

## Referee Comment (RC2) · Anonymous Referee #2 · 9 Nov 2020

Thank you authors and GMD for giving me an opportunity to review this paper, I hope you find this feedback with the intent of improving the manuscript/significance and not just criticism.

The manuscript describes a new finite difference LES code for incompressible flow with a Boussinesq approximation, that allows for nested ("parent/child") grids that can also address "canopy" boundary conditions. A majority of the manuscript is dedicated to evaluating the effect of the p/c choices on a variety of test cases.

**Major revisions / missing ideas:**

1. Overall the algorithm is defined relative to previous versions of the code, which

makes it difficult for a new reader to understand what PALM 6.0 is capable of and how one might reproduce the results. Is the code open source? Are there any reproducibility artifacts with this manuscript?

2. I was confused in many cases about the base method (5th order in space, 3rd-order in time) with all the interpolation caveats ("0th order" or "constant velocity correction") for nested grids. The full set of compromises is not clear, and their impact is argued away in a few sentences.

3. Several tests are defined and the nested grids are tested in a number of different contexts. However, classical grid convergence studies are lacking (usually 1-2 resolutions, 3-4 to identify any trends would be better), and most of the results are "eyeball norm" comparisons between simulations, time- or space-average statistics, and in one case, experiment.

4. There are a number of algorithm compromises that are made for computational, conservation, or accuracy considerations that make it difficult to know when it would be appropriate to apply the code. The authors do discuss identified anomalies and potential causes, which is refreshing, but it is not clear what limitations this might mean for large-scale simulations.

5. Your efficiency numbers are only relative to nested or not. You should clarify the total number of grid points in each simulation, and a metric like "grid points * total time of simulation / wall clock time" is a decent measure of throughput that others can compare to. You never mention what kind of/how many processors and MPI ranks, etc. Scaling with MPI nodes for weak/strong scaling is an important aspect as well.

**Minor revisions / specific suggestions:**

**Clarification / expansion**

A3 - global time step, this should be mentioned up front

Would be good early on to show a picture of C-grid with topography representation,

stair-step "canopy" example

There are several conservative finite difference algorithms that can handle geometry: see "ghost fluid method" and work by Weller & Shaw @ Reading.

P2, 15 "However, only unstructured grid systems allow to take full advantage of spatially variable resolution." - IBM, cut cell, ghost fluid, etc.

P2, 29 - "anterpolation" from child (nested) to parent? I have never heard that term before.

P3, 5-10 - "blunt-obstacle resolving LES" vs. terrain-following approaches. ** You should include a picture making the distinction in your case (stair step on a terrain-following mesh?) Is this a terrain-following code? Mesh for 4.2.2 smooth hill problem?

Note that the child meshes don't move, as in adaptive mesh refinement.

P3, 19-20 - "we are not aware of any research on obstacle-resolving LES employing two-way coupled nesting approach". In this field, there are many in aeronautics and other CFD.

P4, 3-4 - equidistant horizontal spacing? Variable vertical? (How does it line up?)

P5, 10-15 - maybe a picture showing "allowed" and "not allowed" nesting would help? Are nested child regions allowed to "touch" on faces if their resolutions match (or don't)?

P10-11, Fig 3 - should modify to show grid above/below, showing the values that are required to interpolate onto the fine grid

P13, L20 - again, a picture would help explain "canopy-restricted" interpolation

P30, does this test use a vertically-graded mesh? How is the "cube" cut out of the mesh (does it have to land on grid lines, for example?)

P37, L4 - ah, "globally synchronized time step" should be said up front in introduction . . .

P38, L23 - coupling only at the end of time steps? How would child BC's be interpolated in time without "choppy" 0th-order interpolation (which might create time error imprinting)?

P41 L3, "lead to negative values" well-known problem for "positivity preserving limiters" in weather and CFD

**Numerics / testing**

Overall, the nested parent/child problems this really should have a numerical convergence study applied to it (without LES, to make it reproducible). I would be surprised if it is first-order accurate at p/c boundaries, at best, leading to some steep gradients, which may not be interacting well with the LES model and may take many grid cells to "dissipate".

P11 - L5 - why are you doing first-order upwind just to avoid ghost cells? In general, exchanging a few more ghost cells is not that expensive in terms of communication.

P6, mass / vel correction: P7 top L1-15 "According to our tests, $\Delta u_{pre}$ is typically three or four orders of magnitude smaller than the dominant velocity scales of the flow." However what is the order of accuracy of it? The conservation "fix" effectively introduces a discontinuity into grad p, like a dipole in the child domain.

P10, 10-25, couldn't you do a constrained interpolation instead? That is, one that forces any interpolated values to average to the parent (coarse) value to maintain conservation?

P7, 20 - this makes an assumption that production is always high everywhere. What about the classic Blasius flat plate problem with nested child cutting through the turbulent parts? Later on you argue that's not realistic in fully-developed BL turbulence, but your smooth hill example shows how that model is not always true.

P9, L1 - 0th-order interpolation!, L5 - should say "linear" interpolation as this is just for refinement ratio 2

P12, L9-12 - "According to our experience, the conservation properties of an interpolation method are more important than its local accuracy" is entirely problem dependent, and also relates to what happens when you split/place your child meshes in different locations. You could demonstrate this with a few tests.

P14 - 28, "without any obvious" ... how would you quantify this?

Fig 5 - instead of showing values, you might assess "convergence" by comparing to "ref fine" and noting the (quantitative) differences?

P17, L 10 - is this "kink" introduced from error at p/c interface from vertical grid space changes or from low-order interpolation or conservation fixes?

P17, L24 - "different places" are these close to or far from the p/c interface? .48 km vertical vs. 500m would indicate it's outside the child domain?

Fig 6 - very nice spectral analysis, there seems to be very little deviation. But I would like to see 1 more "ref fine" result, as it is not clear if this is a trend.

Fig. 6 - but again, wouldn't it be better to compare to a very-ref-fine result, and just plot the difference in spectrum? What would be an "acceptable" difference in that case?

Fig 7 - what happens if you move the child mesh to a different location, do you see the same result? Or add a second? Or refine everything to allow "more than 1" recirculation cell?

Fig 9 - what is the "tail" at the outflow edge of the child grid? Is that due to velocity conservation corrections?

Fig's 10 / 11 - again, (relative) diff vs. reference solution is more informative perhaps?

Again, Fig 12 - why such a difference between two "reference" solutions? Again, a difference between them would be more informative. And not clear where the "child domains" are? Oh maybe this *is* the entire child domain? How big is the whole domain?

Fig 13-14 - while comparing to wind tunnel is interesting, is it converging with refinement? Maybe an experiment that moves the child domain around, does 1, 2, or 3, nested?

P32, T1 - is this converging with refinement or technique?

**Edits / small items**

"Cyclic" lateral bc's usually are "periodic" boundary conditions?

Fig 5 - is "ref fine" the same resolution as "child 4" or ?? Should be stated clearly

Fig 5 - "squared brackets" should say "angle brackets"?

P17, L 21 - "fetch" - please explain? Maybe "offset" or "shift"?

P18, L34 - "on the order of"? "Superimpose on"

P20 - dangling sentence is hard to find when reading the text

P30, L32 - "become dependent on"?

P31, L4 - "are closest"?

---

## Author Comment (AC1) · 29 Jan 2021

We thank both anonymous referees and Sebastian Giersch for their valuable comments and suggestions. Please find our detailed point-by-point responses below set in black while the referee comments are set in blue. The changes made to the manuscript are visualised in the supplement file "GMD_nesting_BM_SR_AH_diff.pdf". Page and line numbers as well as other numberings given in the referee comments refer to the original manuscript and the numberings given in the author responses, if any, refer to the revised manuscript.

P = page
L = line number

**Anonymous Referee #1:**

The authors developed an LES-LES nested multi-scale system in the PALM model. A key point is to keep unsteady turbulent behaviors in nesting LES-LES models. As far as I know, they developed the two-way coupling scheme in obstacle-resolving LES models for the first time in the world. They carefully evaluated them through the LES-LES numerical experiments on several types of turbulent flows such as NBL, CBL flows, building and hill flows. I strongly recommend publication of this manuscript after the following points are addressed.

Main comments 1. In page2 and line13: The authors mentioned "many numerical solution methods (e.g. finite-element and finite-volume methods)...". Recently, the Lattice Boltzmann Method (LBM) has also come to be regarded as a useful tool and been applied to wind engineering field. Since the LBM has a merit of quite high-speed calculation, it can quickly conduct large-scale wind simulations even for urban areas resolved by a fine grid by massively parallel computing (e.g., Ahmad et al., 2017). Would you comment on the strength of the PALM model based on a finite difference method by comparison with the LBM method?

Ahmad et al.: Large-Eddy Simulation of the Gust Index in an Urban Area Using the Lattice Boltzmann Method, Boundary-Layer Meteorol, DOI 10.1007/s10546-017-0233-6, 2017.

We do recognize the promise of Lattice Boltzmann Method (LBM), and particularly the potential gains it offers in computational efficiency. However, at the moment, we do not have an expert view on the whole picture; different approaches have their own pros and cons. To appreciate their relevance properly, one must be a developer and/or an expert user. The statement in the manuscript relates to the formulation of other CFD solvers relying on the 'classical' continuum approach. We gladly augment the manuscript with a comment acknowledging the promise of new CFD approaches, such as LBM, with appropriate citations.

2. In page3 and line6: The authors mentioned "according to our knowledge, WRF-LES is not applicable to blunt-obstacle resolving LES required for urban turbulence studies". the limitation of the WRF-LES nesting system. Wiersema et al. (2020) developed the WRF-IBM (immersed boundary method) to enable multiscale simulations over highly complex terrain with dynamically downscaled boundary conditions from the meso-scale to the building-scale. Although their approach is one-way nesting system, they successfully simulated turbulent flows in the urban central district by the WRF using the IBM method. The authors should refer the WRF-IBM study and comment on it.

WIERSEMA et al.: Mesoscale to Microscale Simulations over Complex Terrain with the Immersed Boundary Method in the Weather Research and Forecasting Model, Mon.Wea. Rev. (2020) 148 (2): 577–595.

The contribution of Wiersema et al. (2020) is now acknowledged in the text and the original statement modified accordingly.

3. In page13: It is difficult for readers to understand "Canopy-restricted anterpolation". Would you describe the idea of the anterpolation more clearly?

We have described the concept of anterpolation in Sec 3.5 as clearly and thoroughly as we can. We have no idea how it could be made even more clear. The same applies to its canopy-restricted variant described in Sec. 3.6 as clearly as we can.

4. In page26: There seems to be different between the experiments and LES at a downstream position of 1.25H even for a fine reference simulation case. Is this due to the orthogonal grid system?

The difference is due to insufficient resolution as it was shown also in the LES study of Diebold et al. (2013). We have now included in the results from an additional case featuring a nest domain with 1 m resolution, resulting from a two-stage nested run with 4m in the root domain, and 2-m and 1-m grid resolution in the recursively nested child domains. Please see newly made Figures 13, 14 and 15. The 1 m case exhibits a remarkable agreement with the experimental results particularly at 1.25H, which is mainly attributed to the fact that the strong shear layer in the lee of the hill is much better resolved. We think this further highlights the utility of the nesting approach in resolution-demanding flow problems. In addition to this we also provide more quantitative results in the form of the root-mean square error to show the convergence of the results. At this point we would like to make a short remark concerning the step-like representation of the topography. In previous studies we also implemented a cut-cell approach in PALM and compared this against a step-like representation of topography using the smooth hill setup. There we did not find any significant effect on the profiles downstream on the hill top; actually the cut-cell approach showed almost similar profiles at any location with similar deviations. For this reason we are confident that the deviations in the 2-m simulations are attributed to insufficient resolution of the strong shear layer on the lee side of the hill.

Minor comment 5. In page6: the left (1), right (1), south (2), north (2),→west (1), east(1),...?

This left, right denotation is according to the PALM's boundary nomenclature. We added a clarification on this in the text.

6. In page26: Figure 13 and 14→Figures 13 and 14

Corrected.

**Anonymous Referee #2:**

Thank you authors and GMD for giving me an opportunity to review this paper, I hope you find this feedback with the intent of improving the manuscript/significance and not just criticism.

The manuscript describes a new finite difference LES code for incompressible flow with a Boussinesq approximation, that allows for nested ("parent/child") grids that can also address "canopy" boundary conditions. A majority of the manuscript is dedicated to evaluating the effect of the p/c choices on a variety of test cases.

**Major revisions / missing ideas:**

1. Overall the algorithm is defined relative to previous versions of the code, which makes it difficult for a new reader to understand what PALM 6.0 is capable of and how one might reproduce the results. Is the code open source? Are there any reproducibility artifacts with this manuscript?

At the end of section 2 we have included two sentences where we give a brief overview of what is PALM capable of. There, however, we note that this overview is far from being complete and refer to the PALM overview paper of Maronga et al. (2020) instead. This is because we do not want to replicate things that are described in other papers already, which would make the manuscript unacceptably heavy.

To address the reviewers second comment, PALM is an open-source code as indicated in the manuscript in Appendix C. There should be no reproducibility artifacts as the source-code version used for the test cases reported in the article as well as the input files for the test cases are archived and available to anyone as described in the manuscript in Appendix C.

2. I was confused in many cases about the base method (5th order in space, 3rd-order in time) with all the interpolation caveats ("0th order" or "constant velocity correction") for nested grids. The full set of compromises is not clear, and their impact is argued away in a few sentences.

We understand that the description of the interpolation method and conservation considerations is lengthy and quite complicated, but in our view this whole discussion is indeed necessary. This discussion has to address not only the interpolation of the nested boundary conditions but also the interpolation of variables to the flux point within the advection scheme. This tends to make the discussion more complicated. We made a couple of minor adjustments in effort to improve the readability and lessen the risk of confusion.

It is important to understand that we do not consider the choice of zeroth-order interpolation as a compromise but as a means to maximize flux conservation which is extremely important as we do point out in Sec. 3.4 below Eq. (3) and in its last paragraph. One should keep in mind that the boundary conditions interpolated for a nested domain from its parent solution are always inaccurate on the level of the finer nested grid. One should always allow a development zone between a nested boundary and the principal domain of interest. Much more important than the local accuracy of the interpolated boundary conditions is the global conservation.

3. Several tests are defined and the nested grids are tested in a number of different contexts. However, classical grid convergence studies are lacking (usually 1-2 resolutions, 3-4 to identify any trends would be better), and most of the results are "eyeball norm" comparisons between simulations, time- or space-average statistics, and in one case, experiment.

Rigorous grid-convergence studies are beyond the idea of this study. The present work builds upon the earlier development of the non-nested PALM model and the idea here is to only study the nested model results relative to non-nested fine- and coarse-grid reference results. We agree with the referee in that most of the results are "eyeball norm" comparisons as we compared quantitative metrics only in the cube-array case. Therefore we added a quantitative comparison also in the hill case.

4. There are a number of algorithm compromises that are made for computational, conservation, or accuracy considerations that make it difficult to know when it would be appropriate to apply the code. The authors do discuss identified anomalies and potential causes, which is refreshing, but it is not clear what limitations this might mean for large-scale simulations.

Indeed we have observed a number of issues during the implementation and testing phase of the nesting. Many of them we were able to fix, e.g. by minimizing the conservation errors of the interpolation algorithm, while for few we do not have any solution, e.g. for the artificially-induced secondary circulation. However, we think it is important to report these issues and outline hypotheses, which can be proven or rejected in follow-up studies. In fact, we tested the nesting in rather idealized cases, e.g. in a purely convective boundary layer over flat terrain and averaged over a long time to highlight the secondary circulation. This is presumably a rather extreme case. When it comes to more realistic setups with diurnal cycles, realistic terrain surfaces, etc., such long averaging times are not appropriate and possible implications by artificially induced secondary circulations are minimized. We now added a brief discussion in the conclusion section and especially tackled the question of what do we expect in more realistic large-scale simulations.

5. Your efficiency numbers are only relative to nested or not. You should clarify the total number of grid points in each simulation, and a metric like "grid points * total time of simulation / wall clock time" is a decent measure of throughput that others can compare to. You never mention what kind of/how many processors and MPI ranks, etc. Scaling with MPI nodes for weak/strong scaling is an important aspect as well.

The nesting efficiency measures are indeed relative to the non-nested coarse- and fine reference runs. The performance and scalability of the non-nested PALM has been studied and demonstrated elsewhere, e.g. (Maronga et al., 2015), and our idea here is to build upon those earlier works. The performance comparison in Table 2 is extended by adding the following parameters: total number of grid points, number of parallel processes, wall-clock time, and the performance measure CPU-time per grid point per time step. This measure should enable comparisons with other codes.

**Minor revisions / specific suggestions:**

**Clarification / expansion**

A3 - global time step, this should be mentioned up front.

Yes, we moved this from Appendix A to the first paragraph of Sec. 3.1.

Would be good early on to show a picture of C-grid with topography representation, stair-step "canopy" example.

This is already explained and described in Maronga et al, (2015) and we do not want to repeat it in this manuscript to prevent it from becoming excessively large and heavy.

There are several conservative finite difference algorithms that can handle geometry:see "ghost fluid method" and work by Weller & Shaw @ Reading.

Yes, certainly, but in the present work we wish to focus only on the nesting implementation.

P2, 15 "However, only unstructured grid systems allow to take full advantage of spatially variable resolution." - IBM, cut cell, ghost fluid, etc.

Our point here is that structured grid systems do not generally allow very efficient concentration of grid resolution because grid-lines cannot be added locally, instead addition of grid lines always spans throughout the domain. Methods such as Immersed Boundary Method (IBM), cut cell methods, etc are not directly related to this question. Their purpose is to allow modelling of *arbitrary geometries* using structured Cartesian grids.

P2, 29 - "anterpolation" from child (nested) to parent? I have never heard that term before.

The term anterpolation was first introduced by Sullivan et al (1996) although the concept is older, e.g. Clark and Farley (1984). The concept of anterpolation is explained in Sec 3.5. The following sentence is now added: "The term anterpolation was coined by Sullivan et al. (1996) although the concept is older."

P3, 5-10 - "blunt-obstacle resolving LES" vs. terrain-following approaches.  ** You should include a picture making the distinction in your case (stair step on a terrain-following mesh?) Is this a terrain-following code? Mesh for 4.2.2 smooth hill problem?

All geometries are modelled in the stair-step fashion in PALM, also the smooth hill geometry in Sec. 4.2.2. We added the sentence: "The smooth hill geometry is approximated with the grid-following stair-step geometry in PALM due to its orthogonal grid arrangement and its topography description system, see Maronga et al. (2015)" in the first paragraph of Sec. 4.2.2. to make this clear for the readers. We do not think a picture of this is necessary and we try to keep the number of pictures in control.

Note that the child meshes don't move, as in adaptive mesh refinement.

No, they do not move. This should be clear.

P3, 19-20 - "we are not aware of any research on obstacle-resolving LES employing two-way coupled nesting approach". In this field, there are many in aeronautics and other CFD.

This statement is indeed too loose. We restricted it to the ABL-research context by modifying the sentence as: "At current stage, we are not aware of any research on obstacle-resolving LES in the ABL context employing two-way coupled nesting approach."

P4, 3-4 - equidistant horizontal spacing? Variable vertical? (How does it line up?)

This is very straightforward, but leads easily to high aspect ratio grid cells which is not good for accuracy. Therefore the grid stretching is often used only above the boundary layer. With the nesting the stretching is only allowed in the root domain and only above the top boundary of the highest nested domain. This is because the grid-spacing ratio between parent and child must be integer valued as pointed out in Sec. 3.1. The following sentence is added in Sec. 3.1.: "Therefore in nested runs the grid stretching is only allowed in the root domain and only above the top boundary of the highest nested domain."

P5, 10-15 - maybe a picture showing "allowed" and "not allowed" nesting would help? Are nested child regions allowed to "touch" on faces if their resolutions match (or don't)?

We already have 20 pictures in the manuscript (originally 19 plus one added in this revision phase), thus we would not like to add further pictures that are not absolutely necessary. We think that the item list clearly indicates what are the non-allowed situations. From the users' point of view, this information is also found in the PALM documentation. Furthermore, if a user tries to run a setup violating any of these restrictions, the run will be aborted and a descriptive error message is given accordingly. To answer the specific question, child domains are not allowed to touch each other. There must be a margin of at least four child-grid cells in between. We added this into the second item of the Restrictions item list.

P10-11, Fig 3 - should modify to show grid above/below, showing the values that are required to interpolate onto the fine grid

The parent-grid values that are required to interpolate onto the fine grid are shown here as thick arrows. The purpose of this picture is to illustrate the difference of the cases of odd and even grid-spacing ratios when interpolating the staggered velocity components, and to help explain why we ended up using Eq. (4). For this purpose a two-dimensional view is entirely sufficient and showing the third dimension in this picture would add nothing but complexity. We added another picture after this one in order to illustrate the other aspects related to the interpolation.

P13, L20 - again, a picture would help explain "canopy-restricted" interpolation

By the canopy-restricted **anterpolation** we simply mean that the anterpolation is omitted in the lowest part of the domain occupied by the obstacle canopy. We submit that the explanation in Sec. 3.6 is clear and we see no reason to add an additional picture for this purpose. As stated above, we already have 20 pictures in the manuscript, and we would not like to increase this number any more.

P30, does this test use a vertically-graded mesh? How is the "cube" cut out of the mesh (does it have to land on grid lines, for example?)

The mesh in PALM is fully orthogonal (assuming that vertically graded means 'sloped'). Thus, all the cube faces do coincide with the grid planes at the sides of an Arakawa-C grid

box, meaning that a grid cell is either 100% atmosphere or 100% obstacle. Currently, there is no cut-cell method in PALM.

P37, L4 - ah, "globally synchronized time step" should be said up front in introduction…

As stated above, this is now moved from Appendix A to the first paragraph of Sec. 3.1. We think that introduction is not the right place for this, but we do agree that it is better to mention this in the front part of the manuscript rather than in the appendix.

P38, L23 - coupling only at the end of time steps? How would child BC's be interpolated in time without "choppy" 0th-order interpolation (which might create time error imprinting)?

We think this is a good point. So far we have not started studying this, but our starting point will probably be to keep the nest boundary conditions frozen during the whole parent-domain time step and test how it works. We do not really know yet as this is only an outlook to the future work.

P41 L3, "lead to negative values" well-known problem for "positivity preserving limiters"in weather and CFD

Yes, positivity preserving flux limiters are a well known concept, but not relevant here as here we discuss the positivity of the interpolation from parent to child, not the flux computation. Moreover, this discussion only concerns the alternative interpolation method discussed here in Appendix B. Our chosen zeroth-order interpolation method described and discussed in Sec. 3.4 never leads to negative values. We did not even include the alternative method in the final implementation. We just wanted to bring this possibility up in the appendix.

**Numerics / testing**

Overall, the nested parent/child problems this really should have a numerical convergence study applied to it (without LES, to make it reproducible). I would be surprised if it is first-order accurate at p/c boundaries, at best, leading to some steep gradients,which may not be interacting well with the LES model and may take many grid cells to"dissipate".

As stated above, rigorous grid-convergence studies are beyond the idea of this study. The present work builds upon the earlier development of the non-nested PALM model and the idea here is to only study the nested model results relative to non-nested fine- and coarse-grid reference results. Moreover, one should not even expect perfect grid convergence with the formal order of accuracy in the total domain including the root domain and all the child domains, because the boundary conditions interpolated for a child domain will always be less accurate than what would be allowed by the finer grid of the child domain. Instead, one should see the nesting concept more like a sophisticated means to obtain proper boundary conditions for the nested child domains using their parent-domain solutions.

P11 - L5 - why are you doing first-order upwind just to avoid ghost cells? In general,exchanging a few more ghost cells is not that expensive in terms of communication.

There are three layers of ghost cells in PALM because the 5th order Wicker-Skamarock scheme is used over the cyclic boundaries and internal boundaries, i.e. the sub-domain

boundaries due to the domain decomposition. However, the 5th order scheme is not employed in outer boundaries, but instead the stencil is degraded near the outer boundaries as explained in the text. Technically also the nested boundaries are outer boundaries although physically they are not. Changing this feature of PALM would have required complicated intrusions into other parts of the code, and our strategy has been to keep the nesting system as modular as possible and to minimize the number of changes to the rest of the code. Another reason is that the code architecture does not allow to transfer the parent topography/geometry information to the child from outside the nest boundaries (the child's own topography/geometry information covers only the first ghost-cell layer). We do not want to employ any wider finite-difference stencil blindly without knowing if the outer grid points are fluid points or within terrain or a building. Furthermore, the reduction of local accuracy on the nest boundaries is not that critical since the boundary conditions interpolated for a nested domain from its parent solution are in any case always inaccurate on the level of the finer nested grid as stated earlier.

P6, mass / vel correction: P7 top L1-15 "According to our tests, $\Delta u_{pre}$ is typically three or four orders of magnitude smaller than the dominant velocity scales of the flow." However what is the order of accuracy of it? The conservation "fix" effectively introduces a discontinuity into grad p, like a dipole in the child domain.

The pressure solutions in the parent and child domains are independent, i.e. they are not directly coupled so that the pressure solution of the child domain can here be considered independently of the parent. In case there is an overall mass imbalance over the whole boundary of the child domain, this would create a conflict in the pressure equation and the iterative solution algorithm would likely not converge at all. This is avoided by ensuring the mass balance using this fix.

P10, 10-25, couldn't you do a constrained interpolation instead? That is, one that forces any interpolated values to average to the parent (coarse) value to maintain conservation?

The chosen interpolation is reversible, i.e. constrained if we understand the above given definition of concept of constrained correctly. This is explained in this Section 3.4.

P7, 20 - this makes an assumption that production is always high everywhere. What about the classic Blasius flat plate problem with nested child cutting through the turbulent parts? Later on you argue that's not realistic in fully-developed BL turbulence, but your smooth hill example shows how that model is not always true.

Yes, this simplification is based on the assumption that generation minus dissipation of the SGS TKE dominates over its transport. We have tested that the results are not at all sensitive to this simplification. We adjusted the statement in the text to better reflect the outcome of the comparisons. The Blasius flow is irrelevant here as PALM is meant only for turbulent boundary layers, not laminar ones.

P9, L1 - 0th-order interpolation!

We prefer to use zeroth-order, first-order etc. notation in the document. We have now made these notations consistent throughout the manuscript.

P9, L5 - should say "linear" interpolation as this is just for refinement ratio 2

We are not sure if this comment refers to Eq. (4). If so, the lower branch of Eq. (4) is not limited to grid-spacing ratio 2. It applies to all grid-spacing ratios. In case of 2 it happens to be linear interpolation but not for any other values. This is illustrated in Fig. 3 lower part for grid-spacing ratio 4. There all the nodes marked with violet arrows receive this averaged value according to the lower branch of Eq. (4). To eliminate the risk of this kind of confusion, we changed the text on the lower branch of Eq. (4) as follows: "for grid points $i$ between…" -> "for all grid points $i$ between…"

P12, L9-12 - "According to our experience, the conservation properties of an interpolation method are more important than its local accuracy" is entirely problem dependent, and also relates to what happens when you split/place your child meshes in different locations. You could demonstrate this with a few tests.

It is not clear to us what is the aim of this comment. If the purpose is to question our statement on the importance of the conservation properties, let us answer by a hypothetical counter-question: would such a method be acceptable which would only work properly in such problems where the conservation properties happen to be less important?

Local accuracy limitations across the nest boundaries can always be remedied by placing the study area reasonably far away from any inlet or outlet boundary conditions (which are here provided by the nesting system) and such considerations always remain a responsibility of the user. However, conservation violations are far more difficult to remedy and fundamental in nature. Therefore, the nesting algorithm must prioritize conservation.

P14 - 28, "without any obvious"...how would you quantify this?

This sentence describes the visual impression seen in Fig. 4, which in our view appears very apparent. The purpose of this sentence is not to quantify the similarity. Quantitative comparisons of the child and parent solutions and the reference solutions follow after this.

Fig 5 - instead of showing values, you might assess "convergence" by comparing to "ref fine" and noting the (quantitative) differences?

This is a good idea and we also considered it, but chose not to because it would double the number of plots in the image. We also think that the figure sufficiently indicates the convergence of the spectra from the outer towards the inner parts of the child domain.

P17, L 10 - is this "kink" introduced from error at p/c interface from vertical grid space changes or from low-order interpolation or conservation fixes?

The kink is attributed to a mismatch between the parent and the child solution, more precisely between the representation of temperature profiles in the coarse- and fine grid simulation. As we explain in the text, when the near-surface fine-grid resolution is transferred back to the parent, the temperature profile, including stability, is altered and does not necessarily match the state which the subgrid-scale scheme does represent. In other words, the balance between the surface heating and the vertical transport is disturbed after the anterpolation, while the model tends to reestablish this balance in the next time step. This in turn, leads to vertical exchange of heat in the parent, resulting in such kinks.
In this regard, the kink is not attributed to the interpolation or to the conservative fluxes. The kink also appears in vertically nested simulations (where the child and the parent have the same horizontal extensions) where the only interpolation interface (top boundary of the

child domain) is placed far beyond the location of the kink. Here, we want to emphasize that this is not an algorithmic problem but a general problem in two-way nesting when large near-surface gradients occur but are represented differently on the vertical grid. The only way to weaken this kink in such setups is to run parent and child model with identical vertical resolution, though we would like to note that also in this case the near-surface profiles does not necessarily need to match between parent and child since also the horizontal resolution including its effect on the subgrid-scale filtering width plays a role.

P17, L24 - "different places" are these close to or far from the p/c interface? .48 km vertical vs. 500 m would indicate it's outside the child domain?

The distances of the sampling locations from the lateral boundary are now given explicitly, and the wording of the caption of Fig. 6 is also improved.

Fig 6 - very nice spectral analysis, there seems to be very little deviation. But I would like to see 1 more "ref fine" result, as it is not clear if this is a trend.

We are not completely sure what is meant by "1 more "ref fine" result", but we assume that you suggest adding yet a finer-grid reference result to show the trend of the spectra with increasing resolution. We understand that this would be a nice add on, but it is beyond the scope of this study. Our aim here is to show that the nested-domain solution is closer to the fine-grid reference solution than the coarse-grid reference solution is. The aim here is not to study the grid convergence of PALM-solutions as such as it has been addressed in earlier studies, e.g.for CBL in Hellsten and Zilitinkevich (2013). Description of these ideas is now added in the beginning of Sec. 4.

Fig. 6 - but again, wouldn't it be better to compare to a very-ref-fine result, and just plot the difference in spectrum? What would be an "acceptable" difference in that case?

Again, ref + diffs plots would certainly work, but we're convinced that the principal message also becomes evident with the current plots. Ultimately the acceptable level must be defined by the user and the nesting configuration setup accordingly.

Fig 7 - what happens if you move the child mesh to a different location, do you see the same result? Or add a second? Or refine everything to allow "more than 1" recirculation cell?

Yes, moving around the child domain only alters the location of the secondary circulation, meaning that the occurrence and the strength of the circulation is invariant to the exact placement in this case, which is attributed to the flat homogeneous surface as well the cyclic boundary conditions.

Indeed, we have tested this. Adding a second child placed somewhere next to the first child also results in unphysical circulations, but alters slightly the strength and shape of the circulations. This is because the circulations in the parent domain interact and disturb each other, especially if the distance between the child domains is less than the horizontal scales of the secondary circulations, which usually scale with the boundary-layer depth. For child domains that are placed wide apart from each other, however, we do not expect a significant interaction between the circulations.

Fig 9 - what is the "tail" at the outflow edge of the child grid? Is that due to velocity conservation corrections?

No, it is not due to the mass-conservation correction. We have tested this. It is due to a slight conflict of the child solution and its downstream boundary conditions interpolated from the parent solution. The higher the grid-spacing ratio is, the larger the kink becomes as Fig. 9 shows. This conflict is usually rather mild, but it is observable in friction velocity since it is sensitive to even small perturbations.

Fig's 10 / 11 - again, (relative) diff vs. reference solution is more informative perhaps?

Perhaps yes, but as above, we'd like to submit that the current images do serve their purpose.

Again, Fig 12 - why such a difference between two "reference" solutions? Again, a difference between them would be more informative. And not clear where the "child domains" are? Oh maybe this *is* the entire child domain? How big is the whole domain?

The quite large difference between the two reference solutions is due to the insufficient resolution of the coarse-grid reference simulation, mainly because the strong shear layer and thus the relevant physics on the lee side is not well resolved in the 4-m simulation. We disagree with respect to the reviewers comment concerning to show the relative differences between the simulation. Our purpose is to show how beneficial the nesting is in achieving sufficient resolution in the principal area of interest while using coarser resolution further away, rather than show any convergence of the resultsEven with 2-m resolution the results have not been sufficiently converged as the results from the newly presented 1-m nested simulation indicates. The goal here is to demonstrate that the nested child simulations show no significantly different results as the reference simulations show, which we believe is an essential prerequisite to show grid convergence in many flow problems, especially for large-scale simulation setups.

The dimensions and positioning of the nested domains is now given in the text and the caption of Fig. 12. is also made more clear.

Fig 13-14 - while comparing to wind tunnel is interesting, is it converging with refinement? Maybe an experiment that moves the child domain around, does 1, 2, or 3, nested?

We have made a new simulation for the hill case using two stage nesting such that the second nested domain has 1 m grid spacing. The new results are shown and they seem to converge judging from the differences between the 2 m resolution versus 4 m resolution and those between 1 m resolution and 2 m resolution. However, this does not aim to be a proper grid-convergence study. On the other hand, the 1 m resolution results show good agreement with the measurement data. We have also tested how sensitive the results are to the positioning of the nested domain. This was made using only one nested domain. It turned out that there was no significant effect on the profiles along the hill, even when the upstream boundary of the nested domain was placed close to the hill foot, so the results are not sensitive to the placement of the child.

P32, T1 - is this converging with refinement or technique?

The Table 2 (according to the revised numbering) is compiled to illustrate the differences in pressure drag coefficient as explained on P30 L28-30 and to provide evidence that the canopy-restricted (CR) anterpolation strategy remedies the problem of basic two-way coupling.

**Edits / small items**

"Cyclic" lateral bc's usually are "periodic" boundary conditions?

In our opinion these two words have the same meaning in this context. We use the word "cyclic" since it is according to the PALM nomenclature used e.g. in the PALM documentation.

Fig 5 - is "ref fine" the same resolution as "child 4" or ?? Should be stated clearly

Yes, the fine-grid reference case "Ref fine" has the same resolution as all the child domains: child2, child3 and child4. In this case the grid-spacing ratio is increased by lowering the parent-grid resolution and keeping the child resolution fixed. We added the sentence: "All these child domains have the same resolution as the fine-grid reference simulation." in the figure caption.

Fig 5 - "squared brackets" should say "angle brackets"?

Yes, indeed, this is corrected.

P17, L 21 - "fetch" - please explain? Maybe "offset" or "shift"?

According to the MOT dictionary, the word fetch in technical contexts means (among other meanings) "distance swept by wind". This is what we mean. Offset or shift would be misleading words here.

P18, L34 - "on the order of"? "Superimpose on"

"in the order of" has the same meaning as "on the same order of" and seems to be about equally frequently used, so we keep the "in". "...these may superimpose each other,…" is indeed bad wording. This is now replaced by "...these two may become superimposed,..."

P20 - dangling sentence is hard to find when reading the text

Yes, this is true and we apologize for the inconvenience. We believe that no such flaws will exist in the final typeset article.

P30, L32 - "become dependent on"?

Yes, corrected.

P31, L4 - "are closest"?

Yes, corrected.

**Short comment by Sebastian Giersch**

We thank you for this valuable comment. This matter is indeed quite complicated and we have found it difficult to provide a comprehensible description. You are right that it was not

sufficiently clearly explained. In an attempt to clarify this, we added a schematic drawing as a new Fig. 4. Fig. 4 is based on our original hand-drawn sketch which you attached in your comment. We also added some text describing the sequence of operations: the phase 1 operation, which we now denote Transfer to Boundary Plane (TBP), and the phase 2 operation that is the actual interpolation using formulae (3) and (4).

---

## Author Response (AR2)

We thank the anonymous referee for the important comment:

I recommend that the authors describe the idea on the interpolation method and conservation considerations as discussed at the point 2 in the major comments from reviewer #2. This is an important thing.

We now understand that it really is important to present the conservation considerations and the construction of the interpolation method in a much more comprehensible way. Therefore we carefully restructured and largely rewrote Sec. 3.4. We believe that it is now much more comprehensible for readers.

Some minor mistakes were found and corrected also elsewhere in the manuscript.

[revised manuscript text omitted]